# Solving Imperfect-Recall Games via Sum-of-Squares Optimization

**Rui Zheng** [1]  **Ryann Sim** [2]  **Antonios Varvitsiotis** [1 3 4]

## Abstract

Extensive-form games (EFGs) provide a powerful framework for modeling sequential decision making, capturing strategic interaction under imperfect information, chance events, and temporal structure. Most positive algorithmic and theoretical results for EFGs assume perfect recall, where players remember all past information and actions. We study the increasingly relevant setting of imperfect-recall EFGs (IREFGs), where players may forget parts of their history or previously acquired information, and where equilibrium computation is provably hard. We propose sum-of-squares (SOS) hierarchies for computing ex-ante optimal strategies in single-player IREFGs and Nash equilibria in multi-player IREFGs, working over behavioral strategies. Our theoretical results show that (i) these hierarchies converge asymptotically, (ii) under genericity assumptions, the convergence is finite, and (iii) in single-player non-absentminded IREFGs, convergence occurs at a finite level determined by the number of information sets. Finally, we introduce the new classes of (SOS)-concave and (SOS)-monotone IREFGs, and show that in the single-player setting the SOS hierarchy converges at the first level, enabling equilibrium computation with a single semidefinite program (SDP).

## 1. Introduction

Extensive-form games (EFGs) are a central framework for modelling sequential decision-making under imperfect information, with important applications in economics, operations research, and artificial intelligence (Osborne & Rubin-

stein, 1994; Fudenberg & Tirole, 2026). An EFG represents interactions on a tree in which players (or chance) take turns choosing actions. Each sequence of actions defines a history that either ends at a leaf with payoffs or leads to another decision point. This representation is also able to capture imperfect information, where players cannot distinguish between different nodes. EFGs have been used extensively in modern AI, particularly in solving large-scale imperfect information games (Brown & Sandholm, 2018; Bowling et al., 2015; Moravčík et al., 2017; Bakhtin et al., 2022).

A key distinction between classes of EFGs is the ability of players to retain memory. In perfect-recall EFGs, players remember all past information sets and actions, whereas in imperfect-recall EFGs (IREFGs) some information may be lost, such as forgetting which action was taken or even whether a state has been visited before. Recent work has increasingly focused on IREFGs, since they capture a more realistic model of decision-making. IREFGs are used to abstractify large games, letting agents ignore unimportant details (Waugh et al., 2009; Ganzfried & Sandholm, 2014; Brown et al., 2015; Čermák et al., 2017a). They also provide a natural representation for bounded rationality (Lambert et al., 2019), for describing teams of cooperating agents as a single imperfect-recall decision maker (Von Stengel & Koller, 1997; Celli & Gatti, 2018; Zhang et al., 2022), and for designing privacy-preserving or data-restricted agents (Conitzer, 2019; Conitzer & Oesterheld, 2023).

EFGs admit two principal strategy formalisms: mixed strategies, defined as distributions over pure strategies that select an action everywhere in the game tree, and behavioral strategies, which randomize between actions independently at each information set. The canonical solution concept is a Nash equilibrium (NE), a strategy profile from which no player can unilaterally deviate. Perfect recall plays a crucial role in the theoretical guarantees of EFGs: by Kuhn's theorem (Kuhn, 1953), mixed and behavioral strategies are equivalent in perfect recall EFGs, implying the existence of behavioral NEs and enabling efficient algorithms, such as polynomial-time methods for solving two-player zero-sum EFGs (Koller & Megiddo, 1992).

The imperfect-recall setting is far more challenging. The equivalence between mixed and behavioral strategies breaks down, and mixed NE still exist but may be unimplementable

---

[1]Engineering Systems and Design, Singapore University of Technology and Design, Singapore [2]School of Computing, National University of Singapore, Singapore [3]Centre for Quantum Technologies, National University of Singapore, Singapore [4]Archimedes Research Unit, Greece. Correspondence to: Rui Zheng <rui_zheng@mymail.sutd.edu.sg>.

*Proceedings of the 43$^{rd}$ International Conference on Machine Learning*, Seoul, South Korea. PMLR 306, 2026. Copyright 2026 by the author(s).

since they require perfect memory of the game states. Though behavioral NE are more natural, they might not exist, even in two-player zero-sum IREFGs (Wichardt, 2008). Computationally, solving two-player zero-sum IREFGs is NP-hard (Koller & Megiddo, 1992), deciding whether a single-player IREFG achieves a target value is hard (Gimbert et al., 2020; Tewolde et al., 2023), even relaxed equilibrium notions such as EDT and CDT equilibria are hard to compute (Tewolde et al., 2023), and deciding whether a behavioral NE exists is hard (Tewolde et al., 2024).

Many recent hardness results for IREFGs (Gimbert et al., 2020; Tewolde et al., 2023; 2024) rely on the folklore observation that any IREFG over behavioral strategies can be expressed as a polynomial game over a product of simplices (Piccione & Rubinstein, 1997). Importantly, the transformation from an EFG to its polynomial game representation can be carried out in polynomial time. Polynomial games form a key subclass of continuous games, originally studied by (Dresher et al., 2016), in which each player's utility is a polynomial in the decision variables of all players.

On the other hand, there exists well-developed machinery for handling single- and multi-player optimization problems where utilities are polynomial and strategy sets are semi-algebraic. In the single-player case, the Moment-SOS hierarchy (Lasserre, 2001; Parrilo, 2000) applies to polynomial optimization problems of the form $\max f(x)$ subject to $x \in \mathcal{X}$, where $f$ is a polynomial and $\mathcal{X}$ is defined by polynomial equalities and inequalities. Although such problems are NP-hard in general (Motzkin & Straus, 1965), the hierarchy produces a sequence of semidefinite relaxations whose optimal values converge to the true optimum under mild assumptions. Extensions of the Moment-SOS hierarchy have also been developed for the multi-player polynomial optimization setting, applied to polynomially representable supersets of the equilibrium set (Nie & Tang, 2024).

Thus far, the link between IREFGs and polynomial optimization has mostly served to prove hardness results, but we instead use it as a foundation for positive algorithmic results. Specifically, we ask:

> *What guarantees does the Moment-SOS hierarchy provide for solving IREFGs? Are there structured subclasses of IREFGs where it enables tractable computation?*

**Contributions.** By bringing these tools to imperfect-recall games, we obtain a general, provably convergent framework for computing behavioral equilibria in IREFGs. Moreover, we identify structural conditions under which convergence occurs at finite or low levels of the hierarchy, leading to efficient algorithms for broad and natural subclasses. Specifically, our main contributions are:

- For *single-player IREFGs*, the Moment-SOS hierarchy converges asymptotically to the ex-ante optimal value. Under a *genericity* assumption, we show that convergence is instead finite for almost all games. Moreover, in non-absentminded IREFGs (NAM-IREFGs), we show that exact convergence occurs at a level of the hierarchy depending on the number of infosets in the game.

- For *multi-player IREFGs*, we adapt an approach proposed in recent work by (Nie & Tang, 2024). The method requires multiple instantiations of the Moment-SOS hierarchy and converges asymptotically to a behavioral NE, certifying non-existence otherwise. As with the single-player case, for almost all games the convergence is finite. In NAM-IREFGs, we show that a variant of the method generically converges in finite time, requiring only a single instantiation of the hierarchy.

- We define the tractable subclasses of concave and monotone IREFGs, and SOS-certifiable counterparts thereof. In single-player SOS-concave/SOS-monotone IREFGs, we show that the Moment-SOS hierarchy converges at the *first* level, enabling computation of ex-ante optima with a single SDP.

**Related work in IREFGs.** Existing positive results for IREFGs typically rely on restrictions that admit tractable perfect-recall refinements. A-loss recall games (Kaneko & Kline, 1995; Kline, 2002) limit forgetting to past actions, enabling sufficient conditions for behavioral NE to exist and approximation methods in the two-player zero-sum case (Čermák, 2018). NAM-IREFGs, where players always remember previous decision points, can be transformed into equivalent (though exponentially larger) A-loss recall games (Gimbert et al., 2025). Finally, chance-relaxed skew well-formed games are a subclass of IREFGs where counterfactual regret minimization provably minimizes regret (Lanctot et al., 2012; Kroer & Sandholm, 2016).

## 2. Preliminaries

### 2.1. Imperfect Recall Extensive-Form Games (IREFGs)

We first define extensive-form games of imperfect recall. For a more thorough review of standard concepts in EFGs, the reader is referred to (Fudenberg & Tirole, 2026; Osborne & Rubinstein, 1994).

An $n$-player *extensive-form game* $\mathcal{G}$ is a tuple $\mathcal{G} := \langle \mathcal{H}, \mathcal{A}, \mathcal{Z}, \rho, \mathcal{I} \rangle$ where:

- The set $\mathcal{H}$ denotes the states of the game which are decision points for the players. The states $\pi \in \mathcal{H}$ form a tree rooted at an initial state $r \in \mathcal{H}$. We denote terminal nodes in $\mathcal{H}$ by $\mathcal{Z}$. Each nonterminal state $\pi \in \mathcal{H} \setminus \mathcal{Z}$ is associated with a set of *available actions* $\mathcal{A}_\pi$.

- Given $\mathcal{N} = \{1, \ldots, n\}$, the set $\mathcal{N} \cup \{c\}$ denotes the $n + 1$ players of the game. Each state $\pi \in \mathcal{H}$ admits a label $\mathrm{Label}(\pi) \in \mathcal{N} \cup \{c\}$ which denotes the *acting player* at state $\pi$. The letter $c$ denotes a *chance player*, representing exogenous stochasticity. $\mathcal{H}_i \subseteq \mathcal{H}$ denotes the states $\pi \in \mathcal{H}$ with $\mathrm{Label}(\pi) = i$. Each chance node $\pi \in \mathcal{H}_c$ is associated with a fixed distribution $\mathbb{P}_c(\cdot|\pi)$ over $\mathcal{A}_\pi$, denoting the distribution over actions chosen by the chance player at each node.

- For each $i \in \mathcal{N}$, payoff function $\rho_i : \mathcal{Z} \to \mathbb{R}$ specifies the payoff that player $i$ receives if the game ends at terminal state $z \in \mathcal{Z}$.

- The game states $\mathcal{H}$ are partitioned into *information sets* (also called infosets) ascribed to each player, namely $\mathcal{I}_i \in (\mathcal{I}_1, \ldots, \mathcal{I}_n)$. Each information set $I \in \mathcal{I}_i$ encodes groups of nodes that the acting player $i$ cannot distinguish between, so if $\pi_1, \pi_2 \in I$, then $\mathcal{A}_{\pi_1} = \mathcal{A}_{\pi_2}$. We let $\mathcal{A}_I$ denote the shared action set of infoset $I$.

- For notational convenience, we ascribe a singleton information set to each chance node and define $\mathcal{I}_c$ as the collection of these chance node infosets. For each non-terminal node $\pi \in \mathcal{H} \setminus \mathcal{Z}$, we thus define $I_\pi \in (\mathcal{I}_1, \ldots, \mathcal{I}_n) \cup \mathcal{I}_c$ to be the infoset it belongs to.

**Memory.** There are two key distinctions regarding players' memory. A game has *perfect recall* if no player ever forgets their past history, namely, the sequence of information sets visited, actions taken within those information sets, and any information acquired along the way. Formally, for any information set $I \in \mathcal{I}_i$ and any two nodes $\pi_1, \pi_2 \in I$, the sequence of player $i$'s actions from the root $r$ to $\pi_1$ must coincide with the sequence from $r$ to $\pi_2$; otherwise, the player could distinguish between the two nodes. A game is said to have *perfect recall* if this property holds for all players; otherwise, it is said to have *imperfect recall*.

**Strategy formalism.** A *pure strategy* specifies a deterministic action at every information set of a player. A *mixed strategy* is a probability distribution over pure strategies. However, mixed strategies require players to coordinate their actions across information sets, which implicitly assumes memory and therefore conflicts with the imperfect-recall setting. For this reason, IREFGs are most naturally analyzed using *behavioral strategies*, which specify independent randomizations at each information set. Formally, for any information set $I \in \mathcal{I}_i$, let $\Delta(\mathcal{A}_I)$ denote the simplex of probability distributions over the available actions $\mathcal{A}_I$. A *behavioral strategy* for player $i$ is a mapping $\mu_i : \mathcal{I}_i \to \bigcup_{I \in \mathcal{I}_i} \Delta(\mathcal{A}_I)$, assigning to each information set $I$ a distribution $\mu_i(\cdot \mid I) \in \Delta(\mathcal{A}_I)$. The joint behavioral strategy profile for all players is denoted by $\mu := (\mu_i)_{i \in \mathcal{N}}$.

Kuhn's theorem (Kuhn, 1953) establishes that, in games with perfect recall, behavioral and mixed strategies are

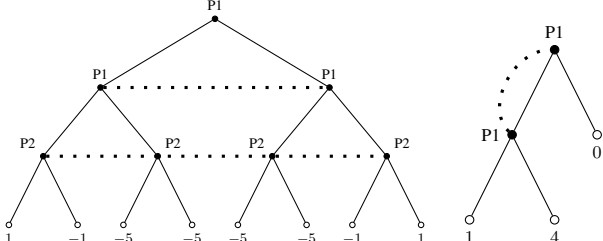

*Figure 1.* (a) A two-player zero-sum IREFG with no behavioral NE; (b) the single-player absentminded taxi driver IREFG. Dotted lines denote infosets. In (b), P1 cannot distinguish between nodes in the same history, so behavioral strategies (e.g., *Left* w.p. $x$, *Right* w.p. $1 - x$) yield expected utility $u(x) = x^2 + 4x(1 - x)$, a polynomial in the behavioral strategy space.

outcome-equivalent—that is, they induce the same probability distribution over terminal histories. This equivalence no longer holds in the imperfect-recall setting, making behavioral strategies the canonical choice for IREFGs.

Going forward, we establish some additional notational conventions regarding behavioral strategies. We use $(\mu_i, \mu_{-i})$ to describe the influence of player $i$ on $\mu$, where $\mu_{-i}$ collects all components except player $i$. Let $\ell_i$ denote the number of infosets of player $i$, i.e. $\ell_i := |\mathcal{I}_i|$, and $m_i^j$ denote the number of actions in a given infoset $I_i^j \in \mathcal{I}_i$ of player $i$, i.e. $m_i^j := |\mathcal{A}_{I_i^j}|$. The strategy set of player $i$ over all their infosets can be written as a Cartesian product of simplices: $\mathcal{S}_i := \bigtimes_{j=1}^{\ell_i} \Delta^{m_i^j - 1}$. Finally, the strategy set over all players is $\mathcal{S} := \bigtimes_{i=1}^{n} \mathcal{S}_i$. In the single-player case, we set $n = 1$ and drop the player index $i$ for clarity.

**Solution concepts.** The expected utility of player $i \in \mathcal{N}$ following (joint) behavioral strategy $\mu$ is $u_i(\mu) := \sum_{z \in \mathcal{Z}} \mathbb{P}(z|\mu, r) \cdot \rho_i(z)$, where $\mathbb{P}(z|\mu, r)$ is the probability that leaf $z \in \mathcal{Z}$ is reached from root $r$ following joint behavioral strategy $\mu$. In a single-player IREFG $\mathcal{G}$, a behavioral strategy $\mu^*$ is said to be *ex-ante optimal* if it is a solution to the following optimization problem:

$$\max_\mu \ u(\mu) \quad \text{s.t.} \ \mu \in \mathcal{S}. \tag{1}$$

In these games, $\mathcal{S}$ is compact and $u$ is continuous, so $\arg\max_\mathcal{S} u \neq \emptyset$ (i.e. ex-ante optima always exist). In a multi-player IREFG $\mathcal{G}$, a joint behavioral strategy $\mu^* \in \mathcal{S}$ is a *(behavioral) Nash equilibrium* if for every player $i \in \mathcal{N}$, we have that:

$$u_i(\mu^*) \geq u_i(\mu_i, \mu_{-i}^*), \quad \forall \mu_i \in \mathcal{S}_i, \tag{2}$$

i.e., no player can profitably deviate from $\mu^*$ to any other behavioral strategy. Importantly, behavioral Nash equilibria need not exist in IREFGs; a counterexample is given in (Wichardt, 2008) (cf. Figure 1 (a)). Moreover, even deciding whether one exists is hard (Tewolde et al., 2024).

## 2.2. The Moment/Sum-of-Squares Hierarchy

We will utilize ideas from the polynomial optimization literature to solve IREFGs. We direct the reader to (Lasserre, 2009b; Laurent, 2008) for more thorough discussions of these techniques. Here we give only a brief overview of the Moment-SOS hierarchy tailored to our purposes; a step-by-step derivation in our setting is provided in Appendix F.

We now introduce the Moment-SOS hierarchy. Consider the polynomial optimization problem:

$$f^* := \max_x \{f(x) : x \in \mathcal{X}\}, \tag{3}$$

where the feasible region is a basic semi-algebraic set $\mathcal{X} := \{x \in \mathbb{R}^m : g_j(x) \geq 0, j \in [\![m_g]\!], h_k(x) = 0, k \in [\![m_h]\!]\}$ for two families of polynomials $g_1, \ldots, g_{m_g}, h_1, \ldots, h_{m_h} \in \mathbb{R}[x]$. Let $\alpha := (\alpha_1, \ldots, \alpha_m) \in \mathbb{N}^m$ be a multi-index, and write $x^\alpha := \prod_{i=1}^m x_i^{\alpha_i}$, $|\alpha| := \sum_{i=1}^m \alpha_i$. For two multi-indices $\alpha, \beta$, the sum $\alpha + \beta$ is taken componentwise, i.e., $(\alpha + \beta)_i = \alpha_i + \beta_i$. Denote by $\mathbb{R}[x]$ the ring of real polynomials in the variables $x := (x_1, \ldots, x_m)$, and by $\mathbb{R}[x]_d$ the vector space of polynomials of degree at most $d$ (whose dimension is $s(m) := \binom{m+d}{m}$). A sequence $y = (y_\alpha)_{\alpha \in \mathbb{N}^m} \subset \mathbb{R}$ defines a Riesz linear functional $L_y : \mathbb{R}[x] \to \mathbb{R}$, $L_y(x^\alpha) = y_\alpha$. Given a sequence $y$ and an integer $d \geq 0$, the *degree-d moment matrix* $M_d(y)$ is the symmetric matrix with rows and columns indexed by all multi-indices $\alpha, \beta$ satisfying $|\alpha|, |\beta| \leq d$, and whose entry is $M_d(y)_{\alpha, \beta} = y_{\alpha + \beta}$. For any polynomial $g(x) = \sum_\gamma g_\gamma x^\gamma$, the *degree-d localizing matrix* of $y$ with respect to $g$ is $M_d(g \star y)_{\alpha, \beta} := L_y(g(x)x^{\alpha+\beta}) = \sum_\gamma g_\gamma y_{\alpha+\beta+\gamma}$, $|\alpha|, |\beta| \leq d - \lceil \deg(g)/2 \rceil$.

Denote by $\Sigma[x] := \{\sigma \in \mathbb{R}[x] : \sigma(x) = \sum_i q_i(x)^2, q_i \in \mathbb{R}[x]\}$ the set of SOS polynomials and by $\Sigma[x]_d$ the set of SOS polynomials of degree at most $2d$. The *quadratic module* of $\mathcal{X}$ is $Q(\mathcal{X}) := \{\sigma_0 + \sum_{j=1}^{m_g} \sigma_j g_j + \sum_{k=1}^{m_h} p_k h_k : \sigma_0, \sigma_j \in \Sigma[x], p_k \in \mathbb{R}[x]\}$. Its truncation at degree $d$, denoted $Q_d(\mathcal{X})$, consists of those elements for which $\deg(\sigma_0) \leq 2d$, $\deg(\sigma_j g_j) \leq 2d$, $\deg(p_k h_k) \leq 2d$. Note that simplex action sets (as are standard in IREFGs) are basic semi-algebraic sets.

Let $d_f := \lceil \deg(f)/2 \rceil$, $d_g := \max_j \lceil \deg(g_j)/2 \rceil$, $d_h := \max_k \lceil \deg(h_k)/2 \rceil$, and $d_\mathcal{X} := \max\{d_g, d_h\}$. For $d \geq d_0 := \max\{d_f, d_\mathcal{X}\}$, the degree-$d$ SOS relaxation of Equation (3) is

$$\begin{aligned} f_d^{\text{sos}} &= \inf_{t, \sigma_0, \{\sigma_j\}, \{p_k\}} t \\ &\text{s.t.} \quad t - f(x) \in Q_d(\mathcal{X}). \end{aligned} \tag{4}$$

Its dual is the degree-$d$ moment relaxation:

$$\begin{aligned} f_d^{\text{mom}} &= \sup_y L_y(f) \\ &\text{s.t.} \quad y_0 = 1, M_d(y) \succeq 0, M_{d-d_\mathcal{X}}(g_j \star y) \succeq 0, \forall j, \\ &\quad L_y(h_k q) = 0, \forall k, \forall q \in \mathbb{R}[x], \deg(h_k q) \leq 2d. \end{aligned} \tag{5}$$

It is immediate that $f_d^{\text{mom}} \leq f_d^{\text{sos}}$ for all $d$, and both sequences $\{f_d^{\text{sos}}\}$ and $\{f_d^{\text{mom}}\}$ are nonincreasing in $d$. Moreover, the hierarchy converges asymptotically under mild assumptions on the description of the semi-algebraic set, e.g., see (Lasserre, 2001). In addition to approximating the optimal value, the moment relaxation also allows one to extract maximizers under appropriate assumptions. It is well known that if, at some order $d$, the flatness condition $\text{rank}\, M_s(y_d) = \text{rank}\, M_{s-d_\mathcal{X}}(y_d)$ holds for some $s \leq d$, then the relaxation is exact and an optimal strategy can be extracted from $M_s(y_d)$; see (Henrion & Lasserre, 2005) and Appendix F.4 for details about the extraction procedure.

# 3. The Link between IREFGs and Polynomial Optimization

We begin this section with a folklore result that bridges the study of IREFGs with polynomial optimization, originating in (Piccione & Rubinstein, 1997) and expanded further in (Tewolde et al., 2023; 2024). A thorough exposition of this connection is given in Appendix C.

**Theorem 3.1** (Folklore). *In any IREFG, the expected utility of a player $i \in \mathcal{N}$ under a joint behavioral strategy $\mu$ is a polynomial in the entries of $\mu$. As a consequence:*

*(i) In the single-player case, computing an ex-ante optimal behavioral strategy reduces to a polynomial optimization problem over products of simplices.*

*(ii) In the multi-player case, computing a behavioral Nash equilibrium $\mu^* \in \mathcal{S}$ reduces to solving coupled polynomial optimization problems over products of simplices, i.e., $\mu_i^* \in \arg\max_{\mu_i \in \mathcal{S}_i} u_i(\mu_i, \mu_{-i}^*), \forall i \in \mathcal{N}$.*

*These reductions can be carried out in polynomial time.*

In view of this, while the Moment-SOS hierarchy applies directly in the single-player case, this is not the case for the multi-player setting. We next introduce two modifications that extend its applicability to the multi-player case and provide stronger convergence guarantees.

**KKT-based Moment-SOS hierarchy.** In the multi-player case, the reduction yields a polynomial game over a product of simplices. However, it is unclear how to use this, since the set of NEs admits no explicit semi-algebraic description. To apply the Moment-SOS hierarchy in this setting, we follow (Nie & Tang, 2024). At a high level, their approach defines a semi-algebraic superset of the Nash equilibria and then applies the Moment-SOS hierarchy to this larger set. This set is obtained by concatenating the KKT conditions of each individual player's optimization problem. These conditions are already polynomial in the primal and dual variables, but under appropriate conditions it is possible to reduce variables by expressing the dual multipliers as polynomials of the primal variables.

To explain this approach, we introduce some additional notation. The action space of each player $i$ is a product of simplices over its information sets, which admits a simple semi-algebraic description $\mathcal{S}_i = \{\mu_i : h_i^j(\mu) = 0, \ g_i^j(\mu) \geq 0, \ \forall j\}$, where $h_i^j(\mu) := \mathbf{1}^\top \mu_i^j - 1$, $g_{i,a}^j(\mu) := \mu_{i,a}^j \ \forall a$, $g_i^j := (g_{i,1}^j, \dots, g_{i,m_i^j}^j)^\top$. For each player $i$ and information set $j$, we define the block gradient $w_i^j(\mu) := \nabla_{\mu_i^j} u_i(\mu) \in \mathbb{R}^{m_i^j}$. Denote by $\nu_i^j \in \mathbb{R}$ and $\lambda_i^j \in \mathbb{R}_{\geq 0}^{m_i^j}$ the Lagrange multipliers associated with the constraints $h_i^j(\mu) = 0$ and $g_i^j(\mu) \geq 0$, respectively. The KKT stationarity condition at an optimizer requires that $w_i^j(\mu) + \nu_i^j \mathbf{1} + \lambda_i^j = 0$ for all $j$. Taking the inner product with $g_i^j(\mu)$, and using $h_i^j(\mu) = 0$ and $g_i^j(\mu)^\top \lambda_i^j = 0$, yields the following explicit (polynomial) expressions for the multipliers: $\nu_i^j(\mu) = -g_i^j(\mu)^\top w_i^j(\mu)$ and $\lambda_i^j(\mu) = -w_i^j(\mu) - \nu_i^j(\mu)\mathbf{1}$. Hence, all Lagrange multipliers are polynomials in $\mu$.

Furthermore, in our setting (i.e., product of simplices), linear independence constraint qualification (LICQ) holds at every feasible point, and hence the KKT conditions are necessary for optimality. Indeed, at any infoset block the constraints are $\mathbf{1}^\top \mu = 1$ and $\mu_a \geq 0$. Thus, the gradients of the active constraints are $\mathbf{1}$ and $e_a$ for all $a$ with $\mu_a = 0$. Since $\sum_a \mu_a = 1$ implies at least one $\mu_a > 0$, not all $e_a$ can be active, so $\{\mathbf{1}\} \cup \{e_a : \mu_a = 0\}$ is linearly independent and LICQ holds. Consequently, every NE must satisfy the joint KKT system of polynomial equations:

$$\begin{cases} w_i^j(\mu) + \nu_i^j(\mu)\mathbf{1} + \lambda_i^j(\mu) = 0, \ \lambda_i^j(\mu) \geq 0, \\ g_i^j(\mu) \geq 0, \ h_i^j(\mu) = 0, \ g_{i,a}^j(\mu)\,\lambda_{i,a}^j(\mu) = 0, \end{cases} \quad \forall i, \, j, \, a.$$
(KKT)

Summarizing, in the multi-player case, and following (Nie & Tang, 2024), a behavioral NE can be computed by applying the Moment-SOS hierarchy to the semi-algebraic set defined by the collection of all coupled (KKT) conditions. We refer to the resulting hierarchy as the *KKT-based hierarchy*.

An important feature of this hierarchy is that it exhibits *finite convergence*, in contrast to the standard asymptotic convergence guarantee. This is achieved under a standard *genericity* assumption on the utility coefficients. More precisely, after fixing the game tree, information structure, and admissible polynomial utility class, the only free parameters are the utility coefficients, denoted collectively by $c$. A property is called *generic* if it holds for all $c$ outside of a proper algebraic variety in this coefficient space. Equivalently, the exceptional set is contained in the solution set of a non-zero system of polynomial equations in $c$. Since every proper algebraic variety has Lebesgue measure zero (Harris, 2013), such a property holds for almost all utility coefficients. We give concrete definitions and additional discussion on the genericity assumption in Appendix D.

For a fixed (multi-)degree, a polynomial is generic if its coefficient vector is generic. (Nie & Tang, 2024, Theorem A.1) show that for a polynomial game with generic utilities and constraints, the KKT equations admit only finitely many complex solutions. Consequently, the corresponding complex variety is finite, and existing results (Laurent, 2008; Lasserre et al., 2008) imply finite convergence of the hierarchy. However, applying this result directly in our setting requires care, since our constraints are not generic but correspond to simplices. Nevertheless, as shown earlier the constraints satisfy LICQ, which turns out to be the only property required in the proof of (Nie & Tang, 2024, Theorem A.1) (i.e., genericity is used there only to imply LICQ).

**Vertex-restricted Moment-SOS hierarchy.** An important structural feature of IREFGs is *absentmindedness*. Absentmindedness refers to the situation where a player cannot distinguish between two nodes that lie along the same history. Formally, a player is *absentminded* if there exist nodes $\pi_1, \pi_2 \in I$ with $\pi_1 \neq \pi_2$ such that $\pi_1$ lies on the unique path from $r$ to $\pi_2$—that is, multiple nodes along the same history belong to the same information set (cf. Figure 1 (b)). A game is *non-absentminded* if no player is absentminded. Whether a player is absentminded has important implications for the corresponding polynomial utility.

**Proposition 3.2.** *In non-absentminded IREFGs (NAM-IREFGs), each player's utility $u_i(\mu)$ is multi-affine in the blocks $\{\mu_i^j = (\mu_{i,a}^j)_{a=1}^{m_i^j}\}_{j=1}^{\ell_i}$, i.e., for any player $i$ and infoset $j$, the map $\mu_i^j \mapsto u_i(\mu)$ is affine when all other blocks $\{\mu_{i'}^{j'}\}_{(i',j')\neq(i,j)}$ are held fixed.*

The proof is given in Appendix H.1. Proposition 3.2 has an interesting consequence for the Moment-SOS hierarchy in the NAM-IREFG case. Indeed, since $u_i$ is affine in each block $\mu_i^j$, and the feasible region is a product of simplices, every blockwise maximization attains its value at an extreme point. More precisely, fixing all variables except $\mu_i^j$ (denote this by $\mu^{-(i,j)}$), the map $\mu_i^j \mapsto u_i(\mu)$ is affine, hence its maximum over $\Delta^{m_i^j}$ is achieved at a vertex $e_{i,a}^j$. Starting from any feasible profile, successively replacing each block $\mu_i^j$ by a maximizing vertex with respect to the current $\mu^{-(i,j)}$ never decreases the objective. After finitely many replacements we obtain a global maximizer whose blocks are all vertices. Defining $b_{i,a}^j(\mu) := \mu_{i,a}^j(\mu_{i,a}^j - 1)$, we see that instead of working with the product of simplices, we can equivalently run the hierarchy over the semi-algebraic vertex-restricted set

$$\mathcal{S}_{i,\mathrm{vr}} := \{\mu_i : h_i^j(\mu) = 0 \ \forall j, \ b_{i,a}^j(\mu) = 0 \ \forall j, a\}, \quad (6)$$

since $\max_{\mu_i \in \mathcal{S}_i} u_i(\mu) = \max_{\mu_i \in \mathcal{S}_{i,\mathrm{vr}}} u_i(\mu)$. We refer to the resulting construction as the *vertex-restricted hierarchy*. In subsequent sections, we explore how this construction enables finite convergence guarantees in the single-player case, and yields computational savings in the multi-player case.

## 4. Single-Player IREFGs

In this section, we focus on *single-player* IREFGs. As explained in the previous section, this setting corresponds to a polynomial optimization problem over a product of simplices, to which the Moment-SOS hierarchy readily applies. We summarize our results for the single-player case below. In addition, Appendix E provides a summary of known complexity results, together with a new hardness result that leverages the connection to POPs.

**Theorem 4.1.** *Consider a single-player IREFG $\mathcal{G}$ with utility function $u$. Let $\ell$ be the number of infosets, $d_0 := \max_{j,a}\{\lceil \deg(u)/2 \rceil, \lceil \deg(g_a^j)/2 \rceil, \lceil \deg(h^j)/2 \rceil\}$, and $u^*$ be the ex-ante optimal value of $\mathcal{G}$. Denote by $u_d^{\mathrm{sos}}, u_d^{\mathrm{sos,kkt}}, u_d^{\mathrm{sos,vr}}$ the values obtained from the Moment-SOS hierarchies applied respectively to the vanilla product-of-simplices, KKT-based, and vertex-restricted formulations. Similarly, we use the superscript* mom *to denote the moment hierarchy. Then we have the following:*

*(i)* $\lim_{d\to\infty} u_d^{\mathrm{sos}} = \lim_{d\to\infty} u_d^{\mathrm{mom}} = u^*$.

*(ii) If $u$ is generic, there exists $d \geq d_0$ with $u_d^{\mathrm{mom,kkt}} = u_d^{\mathrm{sos,kkt}} = u^*$.*

*(iii) If $\mathcal{G}$ is non-absentminded, the degree-$(\ell+1)$ moment relaxation of the vertex-restricted problem is exact: $u_{\ell+1}^{\mathrm{mom,vr}} = u^*$.*

This result capitalizes on the connection between polynomial optimization and IREFGs. While convergence is asymptotic in general, for *almost all* single-player IREFGs, convergence is instead finite. Moreover, in the special case of NAM-IREFGs, we obtain finite convergence at an *explicit level* of the hierarchy which depends on the number of infosets in the game. The proof of this result is deferred to Appendix H.2.

Crucially, we can also *extract* ex-ante optima from the Moment-SOS hierarchies (the full procedure is given in Appendix F.4). In the NAM setting, flatness occurs at a fixed order determined by the number of infosets (Lemma H.1), so extraction is guaranteed at level $\ell + 1$. Hence, under genericity, the KKT-based formulation allows one to extract a solution once finite convergence occurs.

## 5. Multi-Player IREFGs

In this section, we turn our attention to the computation of behavioral NE in *multi-player* IREFGs. Unlike the single-player case, NE are not guaranteed to exist (Wichardt, 2008), and indeed even deciding if one exists is hard (Tewolde et al., 2024). We therefore adopt the *select-verify-cut (SVC)* framework of (Nie & Tang, 2024), which searches for an NE (or certifies nonexistence) by solving a sequence of polynomial subproblems with Moment-SOS relaxations. Specifically,

the SVC method searches for an NE by iterating over three steps:

(1) *Select:* Solve a selector program over the joint (KKT) system (plus any accumulated cuts) to pick an NE candidate.

(2) *Verify:* For each player, fix opponent behavior at the candidate NE and solve a unilateral best-response problem. If no player can improve, the candidate is an NE; otherwise, extract a violated valid inequality from a deviator's best response.

(3) *Cut:* Add new valid inequalities to the selector and re-solve. Each cut removes the non-NE candidate while preserving all NEs.

The method is iterated until a candidate passes verification (NE found) or the selector becomes infeasible (nonexistence certified). A full description of the method is given in Appendix G. Every subproblem in the SVC loop is a polynomial optimization problem, so we can solve them using the Moment-SOS hierarchy. See also (Nie & Tang, 2024, Section 4) for more details.

In general, SVC requires multiple iterations: each round solves one selector Moment-SOS relaxation and up to $n$ verification relaxations (one per player). By contrast, in NAM-IREFGs we can dispense with the verify/cut phases and solve a *single* vertex-restricted Moment-SOS relaxation to compute an NE or certify nonexistence, yielding substantial computational savings. Indeed, since $u_i(\cdot, \mu_{-i})$ is multi-affine in $\mu_i$ (Proposition 3.2), the "no profitable deviation by player $i$" NE condition is equivalent to the family of vertex inequalities $u_i(\mu_i, \mu_{-i}) \geq u_i(v_i, \mu_{-i})$ for all $v_i \in \mathcal{S}_{i,\mathrm{vr}}$. Embedding these inequalities directly into the *Select* step gives the one-shot vertex-restricted program:

$$\min_{\mu} \varphi_\Theta(\mu) := [\mu]_1^\top \Theta [\mu]_1$$
$$\text{s.t.} \begin{cases} \mu \text{ satisfies (KKT)}, \\ u_i(\mu_i, \mu_{-i}) - u_i(v_i, \mu_{-i}) \geq 0, \forall v_i \in \mathcal{S}_{i,\mathrm{vr}}, \forall i, \end{cases} \tag{7}$$

where $[\mu]_1 = (1, \mu^\top)^\top$ and $\Theta \succ 0$ is generic. Feasibility of Equation (7) is equivalent to the existence of a behavioral NE; any feasible point is an NE, and the objective $\varphi_\Theta$ merely selects one among multiple solutions (if any). For instance, replacing $\varphi_\Theta$ with $\max_\mu \sum_{i=1}^n u_i(\mu)$ under the same constraints yields a *welfare-maximizing* NE. Therefore, in the NAM setting, the verify and cut phases are unnecessary.

**Theorem 5.1.** *Let $\mathcal{G}$ be a multi-player IREFG with utility functions $u_i$ for each player $i$. Throughout, subproblems are solved by the KKT-based hierarchies of increasing order. Then, we have the following:*

*(i) The SVC procedure is asymptotically exact: as the relaxation order and number of iterations grow, it returns a behavioral NE when one exists, and otherwise a certificate of nonexistence.*

*(ii) If $u_i$ are all generic, the KKT-based hierarchy has finite convergence for all SVC subproblems, and the SVC loop terminates in finitely many iterations.*

*(iii) If $\mathscr{G}$ is non-absentminded, the Verify/Cut phases in SVC are unnecessary: a single vertex-restricted Select (Equation (7)) suffices to compute an NE or certify nonexistence. Its Moment-SOS hierarchy is asymptotically exact; if $u_i$ are generic, it attains exactness at a finite order.*

The proof is deferred to Appendix H.3. In analogy to the single-player case, Theorem 5.1 shows that finite convergence guarantees can be obtained for *almost all* IREFGs. Under generic utilities, the number of SVC iterations is finite and is at most the number of feasible real KKT candidates, since each failed candidate is permanently excluded by a valid cut. This yields an implicit worst-case bound, though it is combinatorial/algebraic in nature and may still grow rapidly with game size. Comparatively, NAM-IREFGs enjoy significant computational improvements, since only a single Moment-SOS hierarchy suffices for convergence. However, unlike the single-player case, explicit convergence at a fixed game-dependent level is not guaranteed, further highlighting the challenging nature of multi-player IREFGs.

# 6. (SOS)-Concave and (SOS)-Monotone IREFGs

In this section, we focus on defining tractable subclasses of IREFGs, and show how our methods obtain improved convergence guarantees in these subclasses. In the study of continuous games, the seminal work of (Rosen, 1965) introduced concave and monotone games, which exhibit desirable properties—in concave games, NE always exist, and in strictly monotone games, a unique NE exists. Leveraging the connection between IREFGs and polynomials, the following definitions are immediate:

An IREFG $\mathscr{G}$ is *concave* if, for every $i \in \mathcal{N}$, the map $\mu_i \mapsto u_i(\mu_i, \mu_{-i})$ is concave on $\mathcal{S}_i$ for every fixed $\mu_{-i} \in \mathcal{S}_{-i}$. Since $\mathscr{G}$ is polynomial, this is equivalent to the block Hessians being negative semidefinite: $\mathbf{H}_{u_i}(\mu) := \nabla^2_{\mu_i} u_i(\mu) \preceq 0$, $\forall \mu \in \mathcal{S}$, $\forall i \in \mathcal{N}$.

Following (Rosen, 1965), we collect the block partial derivatives into the pseudo-gradient: $v(\mu) := \left(\nabla^\top_{\mu_1} u_1(\mu), \ldots, \nabla^\top_{\mu_n} u_n(\mu)\right)^\top$. Let $\mathbf{J}(\mu) := \nabla v(\mu)$ denote its Jacobian, and define the symmetrized Jacobian $\mathbf{SJ}(\mu) := \frac{1}{2}\left(\mathbf{J}(\mu) + \mathbf{J}(\mu)^\top\right)$. Then, an IREFG $\mathscr{G}$ is *monotone* if $\langle v(\mu) - v(\nu), \mu - \nu \rangle \leq 0$, $\forall \mu, \nu \in \mathcal{S}$. Since $\mathscr{G}$ is polynomial, monotonicity holds if and only if the symmetrized Jacobian of $v$ is negative semidefinite (Rockafellar & Wets, 1998): $\mathbf{SJ}(\mu) \preceq 0$, $\forall \mu \in \mathcal{S}$.

While monotonicity immediately implies concavity, the con-

verse does not hold. Moreover, we can define strict notions of concavity and monotonicity where the block Hessians and symmetrized Jacobian of a game are negative definite, respectively. With these notions in place, we recall the classical existence/uniqueness guarantees (Rosen, 1965, Theorems 1 and 2) tailored to our setting.

**Proposition 6.1.** *Let $\mathcal{S}$ be nonempty, convex, and compact, and let each $u_i$ be continuous.*

*(i) If the IREFG $\mathscr{G}$ is concave, then a behavioral Nash equilibrium exists.*

*(ii) If the IREFG $\mathscr{G}$ is strictly monotone, then the behavioral Nash equilibrium is unique.*

With respect to the SVC method utilized in Section 5, when an IREFG is concave, a joint profile $\mu$ is a Nash equilibrium *if and only if* it is a (KKT) point. Since the strategy set $\mathcal{S}$ is nonempty, convex, and compact, Proposition 6.1 guarantees existence, so the *Select* step is always feasible and returns an NE without any cuts.

**Corollary 6.2.** *If the IREFG $\mathscr{G}$ is concave, then any feasible point of the Select step (without any cuts) is an NE. Furthermore, if $\mathscr{G}$ is strictly monotone, the NE is unique and the Select step returns that unique equilibrium.*

While concavity and monotonicity are widely studied, it is in general hard to *verify* these properties (Ahmadi et al., 2013; Leon et al., 2025). Towards improving the tractability of these classes, (Helton & Nie, 2010) introduced 'effective' SOS-variants thereof, which are verifiable using a single SDP. In particular, we call an IREFG $\mathscr{G}$ *SOS-concave* if, for every $i \in \mathcal{N}$, the negative block Hessian is an SOS-matrix polynomial on $\mathcal{S}$, i.e., there exists a real matrix polynomial $F_i(\mu)$ such that $-\mathbf{H}_{u_i}(\mu) = F_i(\mu)F_i(\mu)^\top$, $\forall \mu \in \mathcal{S}$, $\forall i \in \mathcal{N}$. Furthermore, we call $\mathscr{G}$ *SOS-monotone* if its negative symmetrized Jacobian is an SOS-matrix polynomial, i.e., there exists a real matrix polynomial $F(\mu)$ such that $-\mathbf{SJ}(\mu) = F(\mu)F(\mu)^\top$, $\forall \mu \in \mathcal{S}$.

These are subclasses of concave and monotone IREFGs respectively, so SOS-concavity implies concavity and SOS-monotonicity implies monotonicity. However, the converse does not hold since there exist polynomials which are convex but not SOS-convex (Ahmadi & Parrilo, 2012). Going forward, we explore how these notions can be used to further improve the convergence guarantees of our methods in single-player IREFGs. We note that the definitions of concavity and monotonicity coincide in the single-player case (cf. Proposition H.2). We show that for strict and SOS-concave/monotone single-player IREFGs, our proposed methods obtain stronger convergence guarantees:

**Theorem 6.3.** *Consider a single-player IREFG $\mathscr{G}$ with utility function $u$. Let $d_0 := \max_{j,a}\{\lceil \deg(u)/2 \rceil, \lceil \deg(g_a^j)/2 \rceil, \lceil \deg(h^j)/2 \rceil\}$. Then, the following holds:*

(i) If $\mathcal{G}$ is strictly concave/monotone, then the Moment-SOS hierarchy has finite convergence: there exists $d \geq d_0$ such that $u_d^{\mathrm{sos}} = u_d^{\mathrm{mom}} = u^*$.

(ii) If $\mathcal{G}$ is SOS-concave/SOS-monotone, the degree-$d_0$ Moment-SOS relaxations are exact: $u_{d_0}^{\mathrm{mom}} = u_{d_0}^{\mathrm{sos}} = u^*$, i.e., the Moment-SOS hierarchy converges at the first level.

The proof of the above result is deferred to Appendix H.4. In the case of strictly concave/monotone single-player IREFGs, we obtain a sharper, finite-time guarantee of convergence to the unique ex-ante optimum without needing to modify the vanilla Moment-SOS hierarchy. Moreover, in SOS-concave/SOS-monotone single-player IREFGs, we show that convergence is possible at the first level of the hierarchy, requiring only a single SDP. Furthermore, by leveraging known results (Laurent, 2008; Lasserre, 2009a), once the relaxation is exact, optimal solutions can be obtained directly from the first moments. In particular, for any optimal moment vector $y^*$, $\mu^* = (y_{e_a^j}^*)_{j,a}$. Thus, the extraction procedure as outlined in Appendix F.4 is no longer necessary, enabling efficient extraction of ex-ante optima.

# 7. Experiments

In this section, we present two empirical studies. First, we compare certified global optimization via Moment-SOS with a standard local first-order baseline. Second, we study the empirical scaling behavior of SOS as the number of information sets and the polynomial degree increase. All experiments in this section use randomly generated single-player IREFG instances represented by their polynomial ex-ante utilities over products of simplices. Each game instance has $\ell$ infosets, with 3 actions per infoset, and hence a behavioral strategy space given by a product of $\ell$ probability simplices. For each setting, we generate 50 random utility polynomials $u(\cdot)$ with maximum total degree $D_u$, with coefficients sampled uniformly from $[-1, 1]$. In the SOS implementation, we use the simplex equality constraints to eliminate one redundant variable per information set before constructing the SDP relaxation. Additional examples and implementation details that support the statements in our main theorems, as well as experiments on a generalized absentminded-driver benchmark with multiple infosets and multiple actions, are provided in Appendix I.

## 7.1. SOS Convergence vs. Local Methods

To our knowledge, there is no other general computational baseline for ex-ante global optima or behavioral Nash equilibria in IREFGs. We therefore use projected gradient descent/ascent (PGD) as a local-search baseline since it is known to converge to KKT-type solutions (Fearnley et al., 2022) and was used in recent work focused on local optima

*Table 1.* SOS vs. PGD on single-player IREFGs. $\ell$ is the number of infosets, $D_u$ is the maximum total degree of the utility polynomial, and $D_{\mathrm{SOS}} := 2d$ is the maximum polynomial degree used in the degree-$d$ Moment-SOS relaxation. Results are averaged over 50 instances with 100 PGD restarts per instance. OPT (%) denotes the mean fraction of PGD restarts that reach the SOS-certified global optimum.

| $\ell$ | $D_u$ | $D_{\mathrm{SOS}}$ | SOS time (s) | PGD time (s) | OPT (%) |
|---|---|---|---|---|---|
| 2 | 4 | 4 | 0.02 | 0.05 | 80.72 |
| 2 | 6 | 6 | 0.09 | 0.08 | 88.16 |
| 2 | 8 | 8 | 0.75 | 0.12 | 86.22 |
| 2 | 10 | 10 | 6.64 | 0.18 | 87.14 |
| 3 | 4 | 4 | 0.05 | 0.10 | 79.20 |
| 4 | 4 | 4 | 0.22 | 0.15 | 70.72 |
| 5 | 4 | 4 | 0.88 | 0.18 | 74.56 |
| 6 | 4 | 4 | 3.74 | 0.25 | 72.72 |

in imperfect-recall games (Tewolde et al., 2026). This experiment thus serves to contrast globally-certified optimization via Moment-SOS with a representative local method.

As a local-search baseline, PGD maximizes $u$ over the product of simplices. Let $z \in \mathbb{R}^{3\ell}$ denote the concatenation of all behavioral variables. For each game instance, PGD uses 100 random interior initializations and iterates $z_{t+1} = z_t + \eta \nabla u(z_t)$ with step size $\eta = 0.02$, followed by Euclidean projection onto the feasible region (simplex projection applied independently at each infoset). We terminate when $\|z_{t+1} - z_t\|_2 < 10^{-8}$ or after 5000 iterations, reporting the empirical probability of attaining *global optima*, i.e., the fraction of initializations that reach a global optimum.

The comparison is summarized in Table 1. Across all reported games, SOS *always* recovers an atomic solution, yielding a certificate of global optimality at a relatively modest Moment-SOS order. In our setting, the globally optimal solution is already attained with SOS truncation degree $D_{\mathrm{SOS}} = D_u$, matching the utility polynomial degree. In contrast, PGD requires less time per run for larger games, but is substantially less reliable: even with 100 restarts, the probability of reaching the global optimum is highly instance-dependent. Empirically, some random polynomials are easy (success near $100\%$), while others are hard (success dropping to around $1\%$), indicating that first-order methods can be drawn to suboptimal stationary/KKT points with nontrivial probability. Overall, the Moment-SOS method provides consistent global solutions with certificates, and is competitive with local methods in terms of runtime for smaller game instances.

## 7.2. Scalability of Moment-SOS

The results from the prior experiment indicate that as the game size and level of the hierarchy grows, the runtime of our method increases significantly. We next seek to study the empirical scaling behavior of Moment-SOS on larger

*Table 2.* SOS scalability on larger single-player IREFGs. Each entry reports the average SOS runtime in seconds. Rows vary the number of infosets $\ell$, columns vary the game degree $D_u$, and in all reported entries SOS certifies the optimum with truncation degree $D_{\text{SOS}} = D_u$. Entries marked "–" were not pursued further after the corresponding row or column had already become prohibitively expensive.

| $\ell \backslash D_u$ | 4 | 6 | 8 | 10 | 12 | 14 | 16 |
|---|---|---|---|---|---|---|---|
| 2 | 0.018 | 0.086 | 0.75 | 6.64 | 49.74 | 321.64 | 1674.72 |
| 3 | 0.049 | 1.62 | 54.27 | 1386.38 | – | – | – |
| 4 | 0.22 | 25.05 | 2084.43 | – | – | – | – |
| 5 | 0.88 | 231.80 | – | – | – | – | – |
| 6 | 3.74 | 1694.92 | – | – | – | – | – |
| 7 | 20.90 | – | – | – | – | – | – |
| 8 | 103.92 | – | – | – | – | – | – |
| 9 | 2668.71 | – | – | – | – | – | – |

single-player IREFG instances. Table 2 reports the runtimes of our method in games in which we vary both the number of infosets $\ell$ and the polynomial degree $D_u$. The results indicate that larger domains are solvable, but with rapidly increasing cost. For instance, at fixed $D_u = 6$, the average SOS runtime grows from 0.086s at $\ell = 2$ to 1694.92s at $\ell = 6$. Conversely, at fixed $\ell = 2$, increasing the degree from $D_u = 4$ to $D_u = 16$ increases runtime from 0.018s to 1674.72s. Additionally, the SDP size grows quickly with both the number of behavioral variables and the relaxation order, which practically limits our Moment-SOS approach to IREFGs of modest dimension.

The above observations are in line with known scalability concerns of SOS methods in the literature. Since computational efficiency is not a primary focus of our work, we briefly mention that several recent works have aimed to improve the scalability of SDP solving. For instance, DSOS/SDSOS relaxations trade off solution quality for better computational performance (Ahmadi & Majumdar, 2019), and low-rank SDP methods can reduce memory and runtime by exploiting approximate low-rank structure (Monteiro et al., 2024; Han et al., 2024; 2025; Aguirre et al., 2025). Our polynomial IREFG formulation fits into this framework, so these techniques could be combined with our degree bounds to handle larger imperfect-recall games in future work.

## 8. Discussion and Future Work

Recent work on timeable team games leverages junction-tree/treewidth structure to obtain tractable lifted formulations for team-correlated equilibria (Zhang & Sandholm, 2022; Zhang et al., 2023). Since their setting and solution concept differ from ours, the only directly comparable overlap is the non-absentminded single-player case. A natural direction is to combine our polynomial IREFG formulation with junction-tree variants of Lasserre's hierarchy so the SDP size (and potentially the required level) depends on the constraint-graph treewidth, and to explore whether similar

structure can be leveraged beyond this overlap. We provide a more detailed discussion of this connection in Appendix B.

Future work also includes finding and analyzing other tractable subclasses of IREFGs, and designing faster algorithms that incorporate decentralized methods to compute KKT points. Moreover, while Statement (i) of Theorem 6.3 holds for the Moment-SOS approach, first-order decentralized methods are known to efficiently solve strictly monotone games (see e.g. (Cai & Zheng, 2023; Ba et al., 2025) and references therein). A natural question is how these approaches compare, both theoretically and empirically. Finally, as discussed at the end of Section 7, while we have implemented our methods on some relatively small examples, implementing more tractable SDP solvers to improve scalability remains a crucial direction to explore.

## Acknowledgements

This work is supported by the MOE Tier 2 Grant (MOE-T2EP20223-0018), Ministry of Education Singapore (SRG ESD 2024 174), the CQT++ Core Research Funding Grant (SUTD) (RS-NRCQT-00002), the National Research Foundation Singapore and DSO National Laboratories under the AI Singapore Programme (Award Number: AISG2-RP-2020-016), and partially by Project MIS 5154714 of the National Recovery and Resilience Plan, Greece 2.0, funded by the European Union under the NextGenerationEU Program. The authors also thank anonymous reviewers for their insightful feedback during the review process.

## Impact Statement

This paper presents work whose goal is to advance the fields of Game Theory and Machine Learning. There are many potential societal consequences of our work, none of which we feel must be specifically highlighted here.

## Reproducibility Statement

The code used to generate our experiments use standard Julia scientific computing packages, alongside the SumOfSquares package (Legat et al., 2017; Weisser et al., 2019). Our implementation and scripts for reproducing the experiments are available at `https://github.com/RuiZheng33/Solving-IREFGs-via-SOS`.

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

## A. Additional Related Work

**Sum-of-Squares in Polynomial Games.**    Outside of the works cited in the introduction, few other works have studied IREFGs from the perspective of polynomial optimization. However, games with polynomial utility functions have been proposed and studied in the past (Dresher et al., 2016). (Parrilo, 2006; Laraki & Lasserre, 2012) used semidefinite programming methods to find the value of two-player zero-sum polynomial games, and (Stein et al., 2008) applied similar techniques to separable games (where utilities take a sum-of-products form). Recently, (Nie & Tang, 2023; 2024) also used semidefinite programming techniques to solve for Nash equilibria in $n$-player polynomial games under certain genericity assumptions, or otherwise detect the nonexistence of equilibria. Moreover, (Bach, 2025) studied sum-of-squares relaxations for min-max problems, deriving convergence results of their proposed hierarchies. Finally, (Leon et al., 2025) used sum-of-squares hierarchies to certify concavity and monotonicity in polynomial games.

**Equilibrium Computation in IREFGs.**    Despite the hardness of computing equilibria in IREFGs, some work has studied the special case of two-player, zero-sum IREFGs and derived approximate algorithms to solve them. In particular, (Bosansky et al., 2016; Čermák et al., 2017b) used MILP techniques to solve for minmax-optimal strategies. In the case of non-absentminded two-player zero-sum IREFGs, (Čermák et al., 2018) also used MILP-based methods to obtain scalable algorithms that can approximate minmax strategies. In the case where the IREFGs are timeable and two-player zero-sum (a subclass of non-absentminded two-player zero-sum games), (Zhang & Sandholm, 2022; Zhang et al., 2023) utilized tree-decomposition-based methods to obtain LP/CFR bounds for computing team-correlated (mixed) equilibria.

## B. Relationship to tree-decomposition-based methods

We briefly discuss connections to recent tree-decomposition based algorithms for imperfect-recall team games (Zhang & Sandholm, 2022; Zhang et al., 2023). Their results are not directly comparable to ours in either setting or solution concept: they study two-player zero-sum *timeable* team games (a subclass of non-absentminded games) and compute team-correlated (mixed) equilibria via junction-tree constructions, whereas we study general multi-player IREFGs (including absentminded games) and target *behavioral* equilibria using SOS relaxations.

The primary overlapping game subclass between our settings is the *single-player non-absentminded* "team vs. nature" case, where both approaches reduce to optimizing a multilinear objective over a product of simplices (equivalently, over pure strategies in $\{0, 1\}^n$). In this overlap, the most meaningful comparison is in terms of the *size/structure* of the resulting convex programs (LP versus SDP).

From a treewidth viewpoint, tree-decomposition-based formulations depend on the treewidth $t$ of a dependency graph induced by the constraints. In the canonical simplex formulation, each infoset contributes a normalization constraint $\sum_{a \in \mathcal{A}(I)} \mu_a = 1$, which already enforces $t \geq \max_I |\mathcal{A}(I)| - 1$. More generally, for $0 - 1$ multilinear systems, treewidth-based conditions characterize when Lasserre relaxations become exact (Wainwright & Jordan, 2004). By contrast, our non-absentminded exactness guarantee depends on the number of infosets: Moment-SOS becomes exact at order $\ell + 1$, independent of $\max_I |\mathcal{A}(I)|$. This highlights a regime where our bound can be favorable, namely when infosets have many actions but the information structure is shallow (large $\max_I |\mathcal{A}(I)|$, small $\ell$).

More broadly, these works motivate incorporating treewidth-dependent bounds into our framework, e.g., via sparse/junction-tree variants of Lasserre's hierarchy on the correlative sparsity graph. Extending such structure-exploiting guarantees beyond the single-player non-absentminded overlap (e.g., to absentminded or multi-player behavioral equilibrium constraints) is an interesting direction for future work.

## C. From IREFGs to Polynomials and Back

The equivalence between IREFGs and polynomial optimization is crucial to our proposed methods. In particular, the translation from IREFGs to polynomials is classical (Piccione & Rubinstein, 1997), and we give a full description here for clarity. First, let $P(h'|\mu, h)$ denote the realization probability of reaching $h'$ given that players using strategy $\mu$ are at state $h$. Note that if $h \notin \mathrm{hist}(h')$ (i.e., if $h'$ is not reachable from $h$) then the probability is 0. Intuitively, the realization probability given a behavioral strategy is just the product of choice probabilities along the path from $h$ to $h'$. In order to formally define $P(h'|\mu, h)$, we will need some additional notation. First, any node $h \in \mathcal{H}$ uniquely corresponds to a history $\mathrm{hist}(h)$ from root $r$ to $h$.

- Function $\delta(h) : \mathcal{H} \to \mathbb{N}$ denotes the depth of the game tree starting from node $h \in \mathcal{H}$.

- Function $\nu(h, d) : \mathcal{H} \times \mathbb{N} \to \mathcal{H}$ identifies the node ancestor at depth $d \leq \delta$ from node $h$.

- Function $\alpha(h, d) : \mathcal{H} \times \mathbb{N} \to \cup_{h \in \mathcal{H}} \mathcal{A}(h)$ identifies the action ancestor at depth $d \leq \delta$ from node $h$.

Together, the sequence $(\nu(h, 0), \nu(h, 1), \ldots, \nu(h, \delta(h)))$ uniquely identifies the history of nodes from $r$ to $h$. Likewise, the sequence $(\alpha(h, 0), \alpha(h, 1), \ldots, \alpha(h, \delta(h) - 1))$ uniquely identifies the history of actions taken from $r$ to $h$. Then, the realization probability of node $h'$ from $h$ if the players use joint strategy profile $\mu$ is given by:

**Definition C.1** (Realization Probability).

$$P(h'|\mu, h) = \prod_{j=\delta(h)}^{\delta(h')-1} \mu(\alpha(h', j)|I_{\nu(h', j)}) \quad \text{if } h \in \text{hist}(h').$$

**Definition C.2** (Expected Utility for Player $i$). *For player $i$ at node $h \in \mathcal{H} \setminus \mathcal{Z}$, if strategy profile $\mu$ is played, their expected utility is given by $u_i(\mu|h) := \sum_{z \in \mathcal{Z}} (P(z|\mu, h) \cdot \rho_i(z))$. In its complete form, we can write the expected utility for each player as follows:*

$$u_i(\mu) = \sum_{z \in \mathcal{Z}} \left( \prod_{j=0}^{\delta(z)-1} \mu(\alpha(z, j)|I_{\nu(z, j)}) \cdot \rho_i(z) \right)$$

With some abuse of notation, we can write $P(h|\mu) := P(h|\mu, r)$ where $r$ is the root node, and similarly $u_i(\mu) := u_i(\mu|r)$. Notice that by definition, the expected utility of each player is a polynomial function. In particular, $P(z|\mu, h) \cdot \rho_i(z)$ is a monomial in $\mu$ multiplied by a scalar.

(Tewolde et al., 2023) also recently established that any polynomial can be transformed into a single-player IREFG. Subsequently, (Tewolde et al., 2024) extended this result to a set of polynomials, and multi-player IREFGs. We report the single-player variant of the theorem below, and provide a concrete example to aid readability.

**Theorem C.3.** *Given a polynomial $p : \times_{j=1}^{\ell} \mathbb{R}^{m^j} \to \mathbb{R}$, we can construct a single-player extensive-form game with imperfect recall $\mathscr{G}$ such that the expected utility function of $\mathscr{G}$ satisfies $u(\mu) = p(\mu)$, with $\mu \in \times_{j=1}^{\ell} \Delta^{m^j-1}$. Moreover, the construction can be done in polynomial time.*

*Proof.* The proof follows the analysis of (Tewolde et al., 2023), with minor modifications to notation. Let $d_p = \deg(p)$ and write

$$p(x) = \sum_{D \in \text{MB}(d_p, \boldsymbol{m})} \lambda_D \prod_{j=1}^{\ell} \prod_{a=1}^{m^j} (x_a^j)^{D_a^j},$$

where $\boldsymbol{m} := (m^j)_{j=1}^{\ell}$, $\lambda_D$ are rational, and

$$\text{MB}(d_p, \boldsymbol{m}) := \left\{ D = (D_a^j)_{j,a} \in \times_{j=1}^{\ell} \mathbb{N}_0^{m^j} : \sum_{j=1}^{\ell} \sum_{a=1}^{m^j} D_a^j \leq d_p \right\}.$$

Let $\text{supp}(p) := \{D \in \text{MB}(d_p, \boldsymbol{m}) : \lambda_D \neq 0\}$ and $|D| := \sum_{j,a} D_a^j$. For each $D$, define the multiset $\text{supp}(D)^{\text{ms}}$ that contains $D_a^j$ copies of the pair $(j, a)$ when $D_a^j > 0$. Then $|\text{supp}(D)^{\text{ms}}| = |D|$. The input encoding of $p$ consists of $\ell$, $(m^j)_{j=1}^{\ell}$, and the rational coefficients $(\lambda_D)_{D \in \text{supp}(p)}$.

Given such a polynomial function, we build a single-player extensive-form game $\mathscr{G}$ with imperfect recall whose information sets are $I^j$ ($j \in [\ell]$), each with action set $\mathcal{A}_{I^j} = \{\tau_1, \ldots, \tau_{m^j}\}$. The game $\mathscr{G}$ has a chance root and depth at most $d_p + 1$. The chance node $h_0$ has one outgoing edge to a node $h_D$ for each $D \in \text{supp}(p)$, and chance selects each $h_D$ with probability $1/|\text{supp}(p)|$.

Fix a deterministic ordering $\prec$ of the multiset $\text{supp}(D)^{\text{ms}}$ (e.g., lexicographic on pairs $(j, a)$, repeated $D_a^j$ times). Initialize the *current edge* as the chance edge from $h_0$ into $h_D$.

- If $D = \mathbf{0}$ (the zero multi-index), make $h_D$ terminal with payoff $u(h_D) = \lambda_{\mathbf{0}} \cdot |\operatorname{supp}(p)|$.

- Otherwise, for each next element $(j, a)$ of $\operatorname{supp}(D)^{\mathrm{ms}}$ (in order $\prec$), do:

  - Insert a nonterminal decision node $h$ on the current edge and assign $h$ to the information set $I^j$.
  - Create $m^j$ outgoing edges from $h$, one for each action in $\mathcal{A}_{I^j} = \{\tau_1, \ldots, \tau_{m^j}\}$.
  - For every edge labeled $\tau_{a'}$ with $a' \neq a$, attach a terminal node with utility $0$.
  - Update the *current edge* to be the unique edge labeled $\tau_a$.

After all elements of $\operatorname{supp}(D)^{\mathrm{ms}}$ have been processed, terminate the current edge with a terminal node $z_D$ and set its utility to $u(z_D) = \lambda_D \cdot |\operatorname{supp}(p)|$. This procedure yields a subtree $T_D$ of depth $|\operatorname{supp}(D)^{\mathrm{ms}}| = \sum_{j,a} D_a^j = |D|$.

In this reduction, any point $x = (x_a^j)_{j,a} \in \bigtimes_{j=1}^{\ell} \Delta^{m^j - 1}$ induces a behavioral strategy $\mu$ in $\mathscr{G}$ by $\mu(a_j \mid I^j) = x_a^j$ for all $j, a$. Let $z_D$ denote the terminal node associated with monomial index $D$ (including $D = \mathbf{0}$). At the chance root, $\mathbb{P}(h_D \mid \mu) = 1/|\operatorname{supp}(p)|$ for each $D \in \operatorname{supp}(p)$. If $D = \mathbf{0}$, then $z_D$ is reached immediately, so $\mathbb{P}(z_D \mid \mu) = 1/|\operatorname{supp}(p)|$. If $D \neq \mathbf{0}$, the loop that builds $T_D$ creates exactly $|D|$ decision nodes along the designated branch. At each visit to information set $I^j$, the unique continuing edge is labeled $\tau_a$ and is chosen with probability $x_a^j$. Since $(j, a)$ appears $D_a^j$ times in $\operatorname{supp}(D)^{\mathrm{ms}}$, we obtain

$$\mathbb{P}(z_D \mid \mu) \;=\; \frac{1}{|\operatorname{supp}(p)|} \prod_{j,a=1}^{\ell, m^j} (x_a^j)^{D_a^j}.$$

All sibling edges terminate with utility $0$ and do not contribute.

Therefore the expected utility is

$$
\begin{aligned}
u(\mu) &= \sum_{D \in \operatorname{supp}(p)} \mathbb{P}(z_D \mid \mu) \cdot \rho(z_D) \\
&= \sum_{D \in \operatorname{supp}(p)} \left( \frac{1}{|\operatorname{supp}(p)|} \prod_{j,a} (x_a^j)^{D_a^j} \right) \cdot \lambda_D \cdot |\operatorname{supp}(p)| \\
&= \sum_{D \in \operatorname{supp}(p)} \lambda_D \cdot \prod_{j,a} (x_a^j)^{D_a^j} \\
&= p(x).
\end{aligned}
$$

This extends to $u(\mu) = p(\mu)$ for all $\mu \in \bigtimes_{j=1}^{\ell} \Delta^{m^j - 1}$.

For each $D \in \operatorname{supp}(p)$, $T_D$ contributes a path of length $|D| \leq d_p$ where, at each depth, at most $\max_j m^j$ leaves are created. Hence the total number of nodes and edges is

$$O\Big( \sum_{D \in \operatorname{supp}(p)} \big( |D| + |D| \cdot \max_j m^j \big) \Big) \;\subseteq\; O\big( |\operatorname{supp}(p)| \cdot d_p \cdot \max_j m^j \big).$$

All payoffs are rationals of the form $\lambda_D \cdot |\operatorname{supp}(p)|$, and the chance probabilities are $1/|\operatorname{supp}(p)|$, so labels are computable with bit complexity polynomial in the input size. Thus the game $\mathscr{G}$ is produced in polynomial time in the Turing model. $\quad\square$

Clearly, this construction is not unique. Choosing a different total order on the multiset $\operatorname{supp}(D)^{\mathrm{ms}}$ yields a (potentially) different game tree. All such variants are payoff-equivalent, since the reach probability of $z_D$ depends only on the multiplicities $(D_a^j)$, hence $u(\mu) = p(\mu)$ in every case. For concreteness we fix the lexicographic order. Moreover, we provide a concrete example constructing a (single-player) IREFG from a polynomial.

**Example C.4.** *In this example, we index by (infoset, action) using subscripts to avoid ambiguity: the first subscript denotes the infoset and the second denotes the action (e.g., $x_{12}$). Let $\ell = 2$ with $m_1 = m_2 = 2$, and write $x_{11}, x_{12}$ for $I_1$ and $x_{21}, x_{22}$ for $I_2$. Consider*

$$p(x) = 2 + 3x_{11}x_{21} - 5x_{12}x_{22} + 4x_{21}^2.$$

*Then* $\deg(p) = 2$. *Fix the variable order* $x_{11} \prec x_{12} \prec x_{21} \prec x_{22}$ *and use the induced lexicographic order on multi-indices* $D = (D_{11}, D_{12}, D_{21}, D_{22}) \in \mathbb{N}_0^4$. *For each* $D$, *order the multiset* $\mathrm{supp}(D)^{\mathrm{ms}}$ *by listing the pairs* $(j, a)$ *in lexicographic order with multiplicity. With this convention,*

$$\mathrm{supp}(p) = \left\{ D^{(0)} = \mathbf{0}, \ D^{(1)} = (1,0,1,0), \ D^{(2)} = (0,1,0,1), \ D^{(3)} = (0,0,2,0) \right\}$$

*so* $|\mathrm{supp}(p)| = 4$ *and the lexicographic order is* $D^{(0)} \prec D^{(1)} \prec D^{(2)} \prec D^{(3)}$.

*Then, we construct the game tree as follows:*

**Root (chance).** *Create a chance node* $h_0$ *with four equiprobable edges (1/4 each) to* $h_{D^{(t)}}$, $t = 0, 1, 2, 3$.

**Subtree** $T_{D^{(0)}}$ **(constant term).** *Make* $h_{D^{(0)}}$ *terminal with payoff* $\rho = \lambda_{D^{(0)}} \cdot |\mathrm{supp}(p)| = 2 \cdot 4 = 8$.

**Subtree** $T_{D^{(1)}}$ **for** $x_{11}x_{21}$. *Here* $\mathrm{supp}(D^{(1)})^{\mathrm{ms}} = \{(1,1),(2,1)\}$. *Process in order: (1) insert a node* $h_1 \in I_1$; *create two edges labeled* $x_{11}, x_{12}$; *attach 0 to the* $x_{12}$ *edge; move along* $x_{11}$. *(2) insert a node* $h_2 \in I_2$; *create two edges labeled* $x_{21}, x_{22}$; *attach 0 to the* $x_{22}$ *edge; move along* $x_{21}$. *Terminate with* $z_{D^{(1)}}$ *and payoff* $\rho(z_{D^{(1)}}) = \lambda_{D^{(1)}} \cdot 4 = 12$.

**Subtree** $T_{D^{(2)}}$ **for** $x_{12}x_{22}$. *Here* $\mathrm{supp}(D^{(2)})^{\mathrm{ms}} = \{(1,2),(2,2)\}$. *Process in order: (1) insert* $h_3 \in I_1$; *create* $x_{11}, x_{12}$; *attach 0 to* $x_{11}$; *move along* $x_{12}$. *(2) insert* $h_4 \in I_2$; *create* $x_{21}, x_{22}$; *attach 0 to* $x_{21}$; *move along* $x_{22}$. *Terminate with* $z_{D^{(2)}}$ *and payoff* $\rho(z_{D^{(2)}}) = \lambda_{D^{(2)}} \cdot 4 = -20$.

**Subtree** $T_{D^{(3)}}$ **for** $x_{21}^2$. *Here* $\mathrm{supp}(D^{(3)})^{\mathrm{ms}} = \{(2,1),(2,1)\}$ *(two copies). Process in order: (1) insert* $h_5 \in I_2$; *create* $x_{21}, x_{22}$; *attach 0 to* $x_{22}$; *move along* $x_{21}$. *(2) insert* $h_6 \in I_2$; *again* $x_{21}$ *continues,* $x_{22}$ *gets 0. Terminate with* $z_{D^{(3)}}$ *and payoff* $\rho(z_{D^{(3)}}) = \lambda_{D^{(3)}} \cdot 4 = 16$.

*The constructed game tree is shown in Figure 2.*

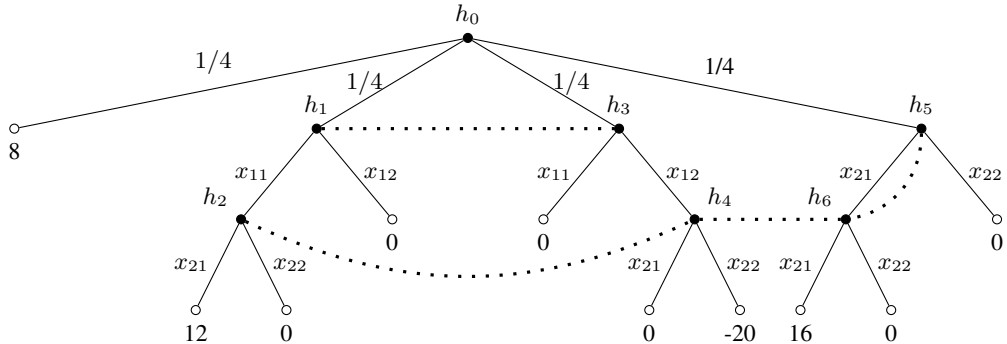

Figure 2. Constructed Single-Player Imperfect-Recall Game for $p(x) = 2 + 3x_{11}x_{21} - 5x_{12}x_{22} + 4x_{21}^2$.

**Verification.** *Given* $x \in \Delta^1 \times \Delta^1$, *define the behavioral strategy by* $\mu(a_j \,|\, I^j) = x_a^j$. *Then*

$$\mathbb{P}(z_{D^{(0)}} \,|\, \mu) = \tfrac{1}{4}, \quad \mathbb{P}(z_{D^{(1)}} \,|\, \mu) = \tfrac{1}{4} x_{11}x_{21}, \quad \mathbb{P}(z_{D^{(2)}} \,|\, \mu) = \tfrac{1}{4} x_{12}x_{22}, \quad \mathbb{P}(z_{D^{(3)}} \,|\, \mu) = \tfrac{1}{4} x_{21}^2.$$

*Hence*

$$
\begin{aligned}
u(\mu) &= \sum_{t=0}^{3} \mathbb{P}(z_{D^{(t)}} \,|\, \mu) \, \rho(z_{D^{(t)}}) \\
&= \tfrac{1}{4} \cdot 8 + \tfrac{1}{4} x_{11}x_{21} \cdot 12 + \tfrac{1}{4} x_{12}x_{22} \cdot (-20) + \tfrac{1}{4} x_{21}^2 \cdot 16 \\
&= 2 + 3x_{11}x_{21} - 5x_{12}x_{22} + 4x_{21}^2 \\
&= p(x).
\end{aligned}
$$

# D. On Genericity

We provide additional clarification on the genericity assumption used in our finite-convergence results. Throughout the paper, genericity is not a condition on the game tree itself. Instead, the game tree, information structure, action sets, and admissible utility class are fixed in advance. The only free parameters are the coefficients of the utility polynomials induced by the terminal payoffs; we denote this coefficient vector by $c$.

A property is called *generic* if it holds for all coefficient vectors $c$ outside a proper algebraic variety in the coefficient space. Equivalently, the exceptional set is contained in the zero set of a nonzero polynomial, or more generally a finite system of polynomial equations, in the entries of $c$. Since every proper algebraic variety has Lebesgue measure zero (Harris, 2013), a generic property holds for almost all utility coefficients. Thus, if the utility coefficients are sampled from any absolutely continuous distribution, the non-generic cases occur with probability zero.

In our setting, for each collection $\alpha$ of active nonnegativity constraints, let $F_\alpha(z; c) = 0$ be the corresponding polynomial KKT system, where $z$ collects the behavioral variables and multipliers. By standard results in elimination theory (Lazard & Rouillier, 2007), there exists a nonzero polynomial $\Delta_\alpha(c)$ such that the exceptional coefficients for this system lie in $\{\Delta_\alpha(c) = 0\}$. We therefore call $c$ generic if $\Delta_\alpha(c) \neq 0$ for every such collection $\alpha$. Equivalently, defining $\Delta(c) := \prod_\alpha \Delta_\alpha(c)$, the nongeneric instances satisfy $\Delta(c) = 0$. Thus the exceptional set is contained in a proper algebraic variety, and hence has Lebesgue measure zero.

Concretely, consider a simple two-player polynomial game:

$$u_1(x, y) = \frac{a}{2}x^2 + bxy + px, \qquad u_2(x, y) = \frac{d}{2}y^2 + exy + qy,$$

with coefficient vector $c = (a, b, p, d, e, q)$. For the interior case, where no nonnegativity constraint is active, the KKT stationarity equations are

$$ax + by + p = 0, \qquad ex + dy + q = 0.$$

The exceptional set is exactly

$$\Delta_\alpha(c) = ad - be = 0.$$

If $ad - be \neq 0$, the system has a unique solution. If $ad - be = 0$, the system may become degenerate. Thus, under any absolutely continuous sampling distribution over the utility coefficients, this exceptional case occurs with probability zero, and the KKT system has finitely many solutions almost surely.

Finally, genericity should be understood relative to the admissible utility class. For example, in a zero-sum game the utilities are not all independent, since they must sum to zero. In that case, one may choose the first $n - 1$ utility functions freely, outside a proper algebraic variety in the corresponding coefficient space, and define the final utility by the zero-sum constraint. Thus our framework applies to generic games within structured classes, including generic zero-sum games, with genericity interpreted after imposing the structural constraints of the class.

# E. On the Computational Complexity of Single-Player IREFGs

While the computational hardness of the ex-ante problem (Equation (1)) has already been established in prior work (Tewolde et al., 2023), we note that one can extend recent results in the polynomial optimization literature to improve these complexity results. First, we introduce a hardness result for finding local optima over polytopes by (Ahmadi & Zhang, 2022):

**Lemma E.1** ((Ahmadi & Zhang, 2022),Theorem 2.6). *Unless $P = NP$, there is no polynomial-time algorithm that finds a point within Euclidean distance $c^n$ (for any constant $c \geq 0$) of a local minimizer of an $n$-variate quadratic function over a polytope.*

**Corollary E.2.** *Finding a local minimizer of a quadratic program over a simplex is NP-hard. Moreover, unless $P = NP$, there is no FPTAS for this problem.*

*Proof.* Specializing the argument in (Ahmadi & Zhang, 2022, Theorem 2.6, p. 7) by replacing $\lceil 3c^n \sqrt{n} \rceil$ by 1 yields the following: unless $P = NP$, there is no polynomial-time algorithm that finds an $\epsilon$-approximate local minimizer (for any constant $\epsilon \in [0, 0.5)$) of a quadratic program over a simplex. If an exact local minimizer could be computed in polynomial time, then, in particular, an $\epsilon$-approximate local minimizer (take $\epsilon = 0$) could also be computed in polynomial time, contradicting Lemma E.1. Therefore, computing an exact local minimizer over a simplex is NP-hard, and no FPTAS exists unless $P = NP$. $\square$

Hence, corresponding to single-player IREFGs, we have:

**Proposition E.3.** *Finding a local optimum of Equation* (1) *is NP-hard. Moreover, unless $P = NP$, there is no FPTAS for this problem. NP-hardness and conditional inapproximability hold even if the game instance $\mathcal{G}$ has no chance nodes, a tree depth of 2, and only one information set.*

Following this, we can improve the results from (Tewolde et al., 2023, Proposition 4):

**Proposition E.4.** *Finding the ex-ante optimal strategy $\mu^*$ of a given extensive-form game instance $\mathcal{G}$ is NP-hard. Specifically:*

1. *Unless $P = NP$, no FPTAS exists. NP-hardness and conditional inapproximability hold even if the game has no chance nodes, a tree depth of 2, and a single information set.*

2. *Unless $NP = ZPP$, no FPTAS exists. NP-hardness and conditional inapproximability hold even if the game has a tree depth of 3 and a single information set.*

3. *NP-hardness holds even without absentmindedness, with tree depth 4 and two actions per information set.*

4. *NP-hardness holds even without absentmindedness, with tree depth 3 and three actions per information set.*

*Proof.* To prove statement 1, we note that since any global optimum is in particular a local optimum, finding a global maximum must be at least as hard as finding a local one. Thus, the same hardness and inapproximability from Proposition E.3 immediately carry over to the problem of finding a global optimum, i.e., ex-ante optimal strategy $\mu^*$.

The statements 2-4 are proven in (Tewolde et al., 2023, Proposition 4). □

Following (Tewolde et al., 2023), we compile the core correspondences between the polynomial optimization formulation of Equation (1) and solution concepts in single-player IREFGs, together with the computational complexity of computing each notion, in Table 3.

*Table 3.* Correspondence between single-player IREFGs and POPs with complexity results.

| POP Optimality Notion | IREFG Equilibrium Notion | | Rel. | Complexity |
|---|---|---|---|---|
| Global maximizer | Ex-ante optimal strategy | | $\Leftrightarrow$ | NP-hard |
| | (EDT,GDH) | with absentmindedness | $\Rightarrow$ | |
| KKT point | | without absentmindedness | $\Leftrightarrow$ | CLS-hard |
| | (CDT,GT) | | $\Leftrightarrow$ | |

**Remark E.5.** *The three equilibrium notions for single-player IREFGs form an inclusion chain in general (Tewolde et al., 2023, Lemma 17): ex-ante optimal $\Rightarrow$ (EDT,GDH)-equilibrium $\Rightarrow$ (CDT,GT)-equilibrium. In games without absentmindedness, this strengthens to an equivalence (Tewolde et al., 2023, Lemma 13): a strategy is (CDT,GT)-equilibrium if and only if it is (EDT,GDH)-equilibrium. Moreover, when the game has a single information set, the blockwise argmax condition of (Tewolde et al., 2023, Lemma 15) reduces to a global argmax, so every (EDT,GDH)-equilibrium is also ex-ante optimal. Since ex-ante optimization is NP-hard, it follows that computing an (EDT,GDH)-equilibrium is already NP-hard in this one-infoset setting.*

## F. The Moment-SOS Hierarchy

In this section, we present a thorough derivation of the Moment-SOS hierarchy which we have specialized to single-player IREFGs.

### F.1. Problem Reformulation

Let $m = \sum_{j=1}^{\ell} m^j$. Denote the strategy variables collectively by $\mu = (\mu_a^j) \in \mathbb{R}^m$. The feasible set is

$$\mathcal{S} = \underset{j=1}{\overset{\ell}{\times}} \Delta(A_{I^j}) = \{\mu \in \mathbb{R}^m : \mu_a^j \geq 0 \ \forall j, a, \ \sum_{a=1}^{m_j} \mu_a^j = 1 \ \forall j\}. \tag{8}$$

Define constraint polynomials

$$g_a^j(\mu) = \mu_a^j, \quad h^j(\mu) = \sum_{a=1}^{m^j} \mu_a^j - 1. \tag{9}$$

By construction,

$$\mathcal{S} = \{\, \mu \in \mathbb{R}^m : g_a^j(\mu) \geq 0, j \in [\![\ell]\!], a \in [\![m^j]\!], \ h^j(\mu) = 0, j \in [\![\ell]\!] \,\}. \tag{10}$$

Accordingly, the quadratic module of $\mathcal{S}$ is

$$Q(\mathcal{S}) = \Big\{ \sigma_0(\mu) + \sum_{j=1}^{\ell} \sum_{a=1}^{m^j} \sigma_a^j(\mu) \, g_a^j(\mu) + \sum_{j=1}^{\ell} p^j(\mu) \, h^j(\mu) : \sigma_0, \sigma_a^j \in \Sigma[\mu], \ p^j \in \mathbb{R}[\mu] \Big\}. \tag{11}$$

An important property in our formulation is the Archimedean property, defined below:

**Definition F.1** (Archimedean property). *The quadratic module $Q(\mathcal{X})$ is* Archimedean *if there exists $N > 0$ such that*

$$N - \|x\|^2 \ \in \ Q(\mathcal{X}).$$

This property guarantees that $\mathcal{X}$ is compact.

For any $\mu \in \mathcal{S}$ one has $0 \leq \mu_a^j \leq 1$, hence $\|\mu\|^2 = \sum_{j,a}(\mu_a^j)^2 \leq \sum_{j,a} \mu_a^j = \sum_{j=1}^{\ell} \sum_{a=1}^{m^j} \mu_a^j = \ell$. Therefore the quadratic polynomial $g_0(\mu) := \ell - \|\mu\|^2$ is nonnegative on $\mathcal{S}$. Adding the redundant inequality $g_0(\mu) \geq 0$ to the description of $\mathcal{S}$ gives $\mathcal{S} = \{\mu : g_a^j(\mu) \geq 0, h^j(\mu) = 0, g_0(\mu) \geq 0\}$, so that $N - \|\mu\|^2 \in Q(\mathcal{S})$ with $N = \ell$. Hence, the quadratic module $Q(\mathcal{S})$ is Archimedean.

From Equation (1), the expected payoff of a single-player IREFG can be written as a polynomial $u(\mu)$. We seek

$$u^* = \max_{\mu \in \mathcal{S}} u(\mu). \tag{12}$$

### F.2. Moment-SOS Relaxation

To approximate the nonconvex problem in Equation (12), we use Lasserre's Moment-SOS hierarchy. We derive a pair of dual hierarchies—one in the space of SOS multipliers (primal) and one in the space of moments (dual)—which yield provable upper bounds on $u^*$.

**(a) Primal (SOS-relaxation)**

Observe that

$$u^* = \inf\{t : t - u(\mu) \geq 0 \quad \forall \mu \in \mathcal{S}\} = \inf\{t : t - u(\mu) > 0 \quad \forall \mu \in \mathcal{S}\}. \tag{13}$$

Since $\mathcal{S} = \{\mu : g_a^j(\mu) \geq 0, \ h^j(\mu) = 0\}$ has Archimedean quadratic module $Q(\mathcal{S})$, Putinar's Positivstellensatz (Putinar, 1993) gives the equivalent infinite-dimensional certificate

$$u^* = \inf\{t : t - u(\mu) \in Q(\mathcal{S})\}. \tag{14}$$

However, membership in $Q(\mathcal{S})$ is still an infinite-dimensional constraint.

**Truncation to finite SDP.** Let $d_u := \lceil \deg(u)/2 \rceil$ and $d_{\mathcal{S}} := \max_{j,a}\{\lceil \deg(g_a^j)/2 \rceil, \lceil \deg(h^j)/2 \rceil\} = 1$, then $d_0 := \max\{d_u, d_{\mathcal{S}}\} = d_u$. Fix any relaxation order $d \geq d_0$. We truncate the sums-of-squares and polynomial multipliers to degree $2d$, obtaining:

$$\begin{aligned} u_d^{\mathrm{sos}} = \inf_{t, \sigma_0, \{\sigma_a^j\}, \{p^j\}} \ & t \\ \text{s.t.} \quad & t - u(\mu) \in Q_d(\mathcal{S}). \end{aligned} \tag{15}$$

**(b) Dual (Moment-relaxation)**

We begin by expressing the original problem (12) in the space of Borel measures $\phi$ supported on $\mathcal{S}$. Since any admissible $\phi$ must satisfy $\phi \geq 0$ and $\int d\phi = 1$, one has the exact infinite-dimensional program

$$u^* = \sup_{\substack{\phi \in \mathcal{M}_+(\mathcal{S}) \\ \int d\phi = 1}} \int u(\mu)\, d\phi(\mu). \tag{16}$$

A measure $\phi$ is equivalently described by its full sequence of moments

$$y_\alpha = \int \mu^\alpha\, d\phi(\mu), \quad \forall \alpha \in \mathbb{N}^m. \tag{17}$$

Introduce the Riesz functional $L_y : \mathbb{R}[\mu] \to \mathbb{R}$ by $L_y(\mu^\alpha) = y_\alpha$, $\forall \alpha$. Expand $u(\mu) = \sum_\alpha u_\alpha \mu^\alpha$, so that

$$\int u(\mu)\, d\phi(\mu) = \sum_\alpha u_\alpha \int \mu^\alpha\, d\phi = \sum_\alpha u_\alpha y_\alpha =: L_y(u). \tag{18}$$

Requiring $\phi \geq 0$ and $\mathrm{supp}(\phi) \subseteq \mathcal{S}$ is equivalent to the following linear matrix constraints on $y$:

$$M_d(y) \succeq 0, \quad \forall d \in \mathbb{N} \iff \int v(\mu)^2\, d\phi \geq 0, \quad \forall v \in \mathbb{R}[\mu], \tag{19}$$

$$M_d(g_a^j \star y) \succeq 0, \quad \forall d \in \mathbb{N} \iff \int g_a^j(\mu)\, v(\mu)^2\, d\phi \geq 0, \quad \forall v \in \mathbb{R}[\mu], \tag{20}$$

$$L_y(h^j q) = 0, \quad \forall q \in \mathbb{R}[\mu], \iff \int h^j(\mu)\, q(\mu)\, d\phi = 0, \quad \forall q \in \mathbb{R}[\mu], \tag{21}$$

$$y_0 = 1 \iff \int 1\, d\phi = 1. \tag{22}$$

Thus Equation (16) can be rewritten as the (infinite-dimensional) moment program

$$u^* = \sup_y L_y(u)$$
$$\text{s.t.} \quad (19),(20),(21),(22). \tag{23}$$

**Truncation to finite SDP.** Fix any relaxation order $d \geq d_0$. In practice, we truncate Equation (23) to the degree-$d$ moment relaxation:

$$u_d^{\mathrm{mom}} = \sup_y L_y(u)$$
$$\text{s.t.} \quad M_d(y) \succeq 0,$$
$$M_{d-1}(g_a^j \star y) \succeq 0, \quad \forall j, a,$$
$$L_y(h^j q) = 0, \quad \forall j, \forall q \in \mathbb{R}[\mu],\ \deg(h^j q) \leq 2d,$$
$$y_0 = 1. \tag{24}$$

## F.3. Moment-SOS Hierarchy for Non-Absentminded Games

Recall that because $u$ is multi-affine, we have the following:

$$u^* = \max_{\mu \in \mathcal{S}} u(\mu) = \max_{\mu \in \mathcal{S}_{\mathrm{vr}}} u(\mu). \tag{25}$$

Let $d_0^{\mathrm{vr}} := \max\{d_u, d_{\mathcal{S}_{\mathrm{vr}}}\} = d_u$, and fix any relaxation order $d \geq d_0^{\mathrm{vr}}$. The degree-$d$ SOS relaxation of Equation (25) is

$$u_d^{\mathrm{sos,vr}} = \inf\ t$$
$$\text{s.t.} \quad t - u(\mu) \in Q_d(\mathcal{S}_{\mathrm{vr}}). \tag{26}$$

The corresponding degree-$d$ moment relaxation reads

$$
\begin{aligned}
u_d^{\text{mom,vr}} \;=\; \sup_y \;\; & L_y(u) \\
\text{s.t.} \quad & M_d(y) \;\succeq\; 0, \\
& L_y\big(h^j\, q\big) = 0 \quad \forall j,\ \forall q \in \mathbb{R}[\mu],\ \deg(h^j q) \le 2d, \\
& L_y\big(b_a^j\, q\big) = 0 \quad \forall j,a,\ \forall q \in \mathbb{R}[\mu],\ \deg(b_a^j q) \le 2d, \\
& y_0 = 1.
\end{aligned}
\tag{27}
$$

Note that there are no localizing PSD constraints for $\mu_a^j \ge 0$, because the binomials $b_a^j = 0$ already enforce $\mu_a^j \in \{0,1\} \subset [0,1]$.

### F.4. Pseudo-Expectations and Extracting Solutions

A feasible point $y = (y_\alpha)_{|\alpha| \le 2d}$ of the truncated moment SDP in Equation (24) defines a linear functional $\widetilde{\mathbb{E}}_d : \mathbb{R}[\mu]_{2d} \to \mathbb{R}$ via $\widetilde{\mathbb{E}}_d[\mu^\alpha] = y_\alpha$. This functional behaves like an expectation operator up to degree $2d$: it is normalized ($\widetilde{\mathbb{E}}_d[1] = 1$), positive on squares ($\widetilde{\mathbb{E}}_d[q^2] \ge 0$ for all $q$ with $\deg q \le d$), and it enforces feasibility through the linear identities induced by the constraints. Such a functional is often called a degree-$2d$ pseudo-expectation.

Expanding $u(\mu) = \sum_\alpha u_\alpha\,\mu^\alpha$, the moment objective is $L_y(u) = \sum_\alpha u_\alpha\, y_\alpha = \widetilde{\mathbb{E}}_d[u(\mu)]$. Hence the truncated moment relaxation in Equation (24) can be viewed as:

$$
f_d^{\text{mom}} \;=\; \sup\Big\{\, \widetilde{\mathbb{E}}_d\big[u(\mu)\big] \;:\; \widetilde{\mathbb{E}}_d \text{ is a degree-}2d \text{ pseudo-expectation consistent with } (g \ge 0,\ h = 0)\,\Big\}.
$$

In words, the moment SDP maximizes the pseudo-expected ex-ante payoff over all degree-$2d$ "virtual laws" that satisfy the polynomial feasibility conditions up to degree $2d$.

A feasible $y$ in Equation (24) need not come from any genuine probability measure on $\mathcal{S}$, and it generally encodes only a pseudo-expectation. A fundamental exception is the flat extension condition (Curto & Fialkow, 1996): if for some $s \le d$,

$$
\operatorname{rank} M_s(y) = \operatorname{rank} M_{s-1}(y),
\tag{28}
$$

then there exist atoms $\mu^{(1)}, \dots, \mu^{(r)} \in \mathcal{S}$ with $r = \operatorname{rank}(M_s(y))$ and weights $\lambda_k \ge 0$ with $\sum_k \lambda_k = 1$ such that

$$
\widetilde{\mathbb{E}}_s[p] \;=\; L_y(p) \;=\; \sum_{k=1}^{r} \lambda_k\, p\big(\mu^{(k)}\big) \qquad \forall p \in \mathbb{R}[\mu]_{2s}.
$$

Thus a flat pseudo-expectation is the true expectation with respect to a finitely atomic probability measure supported on $\mathcal{S}$. Consequently, in the flat regime the SDP objective $\widetilde{\mathbb{E}}_s[u] = L_y(u)$ equals the true ex-ante payoff under optimal strategies, and the atoms $\{\mu^{(k)}\}$ (optimal solutions) can be extracted from $M_s(y)$ by standard linear-algebraic procedures (e.g., multiplication matrices).

#### F.4.1. EXTRACTION PROCEDURE

A standard sufficient flatness test certifying exactness and enabling solution extraction is $\operatorname{rank} M_s(y^*) = \operatorname{rank} M_{s-d_K}(y^*)$, where $d_K = d_{\mathcal{S}}$ in Equation (24), and $d_K = d_{\mathcal{S}_{\text{vr}}}$ in Equation (27). Note that $d_{\mathcal{S}} = d_{\mathcal{S}_{\text{vr}}} = 1$, then in both cases the flatness condition specializes to Equation (28).

Let $y^*$ be optimal for the degree-$d$ moment SDP and assume Equation (28) holds at some order $s \le d$. Set $r := \operatorname{rank} M_s(y^*) = \operatorname{rank} M_{s-1}(y^*)$. Then $y^*$ admits an $r$-atomic representing measure $\sum_{k=1}^r \lambda_k\, \delta_{\mu^{(k)}}$ supported on the feasible set, with $\lambda_k > 0$ and $\sum_k \lambda_k = 1$. Since $L_{y^*}(u) = \sum_k \lambda_k u(\mu^{(k)})$ equals the global optimum, every atom satisfies $u(\mu^{(k)}) = u^*$ and hence is a global maximizer of $u$.

We now recover $\{\mu^{(k)}\}_{k=1}^r$ directly from the optimal moment matrix $M_s(y^*)$ by the standard multiplication-matrix routine. The extraction steps are based on (Henrion & Lasserre, 2005).

Let $v_s(\mu)$ be the vector of all monomials in $\mu$ of total degree $\leq s$, with length $N_s = \binom{m+s}{s}$. Then the order-$s$ moment matrix can be represented as:

$$M_s(y^*) \;=\; \sum_{k=1}^{r^*} \lambda_k \, v_s\big(\mu^{(k)}\big) \, v_s\big(\mu^{(k)}\big)^\top \;=\; V^*(V^*)^\top, \tag{29}$$

where $\lambda_k \geq 0$, $\sum_{k=1}^{r} \lambda_k = 1$, and $V^* \in \mathbb{R}^{N_s \times r}$ collects the columns $\sqrt{\lambda_k}\, v_s\big(\mu^{(k)}\big)$.

For computation, we form a rank factor of $M_s(y^*)$ by retaining the $r$ positive modes (e.g., via eigendecomposition or a Cholesky-type factorization):

$$M_s(y^*) \;=\; VV^\top, \qquad V \in \mathbb{R}^{N_s \times r}. \tag{30}$$

By construction, $\mathrm{span}(V) = \mathrm{span}(V^*)$. Hence the columns of $V$ are linear combinations of $\{\sqrt{\lambda_j}\, v_s(\mu^{(k)})\}_{k=1}^{r^*}$.

To obtain an explicit monomial basis of that subspace, we reduce $V$ to column-echelon form by Gaussian elimination with column pivoting, and rescale so that the pivot block is the identity. This gives $\widehat{V} = VT$, $\widehat{V}_{B,:} = I_{r^*}$, with $T \in \mathbb{R}^{r \times r}$ invertible and $B = \{\beta_1, \ldots, \beta_r\}$ the indices of the pivot rows (monomials). The pivot indices select a monomial "generating basis":

$$w(\mu) \;:=\; [\mu^{\beta_1} \;\cdots\; \mu^{\beta_r}]^T.$$

The same elimination step simultaneously produces linear reduction rules for all monomials of degree $\leq s$: each $\mu^\alpha$ is written, on the support, as a linear combination of the generators $\{\mu^{\beta_k}\}_{k=1}^r$. Stacking these relations row-wise yields the rewriting matrix $\mathcal{R} \in \mathbb{R}^{N_s \times r}$ of the form (identity in the pivot rows and coefficients elsewhere)

$$v_s(\mu) \;=\; \mathcal{R}\, w(\mu), \qquad \mathcal{R}_{B,:} = I_r, \tag{31}$$

which is exactly the coordinate change from the standard monomial vector $v_s$ to the generating basis $w$ on the atoms $\{\mu^{(k)}\}_{k=1}^r$.

In the basis $w$, multiplication by each coordinate $\mu_\eta$ acts linearly. For $\eta = 1, \ldots, m$ we build the multiplication matrices $N_\eta \in \mathbb{R}^{r \times r}$ defined by

$$N_\eta\, w(\mu) \;=\; \mu_\eta\, w(\mu). \tag{32}$$

Concretely, for the $k$th basis monomial $\mu^{\beta_k}$, form $\gamma = \beta_k + e_\eta$; if $\gamma \in B$ set $N_\eta(:,k) = e_{\mathrm{row}(\gamma)}$, otherwise take $N_\eta(:,k) = \mathcal{R}_{\gamma,:}^\top$ from Equation (31).

The atoms appear as common eigenpairs of $\{N_\eta\}$. Indeed, with $e_k := w(\mu^{(k)})$ one has $N_\eta e_k = \mu_\eta^{(k)} e_k$ for all $\eta$. For robust computation we form a random convex combination

$$N \;=\; \sum_{\eta=1}^{m} \lambda_\eta N_\eta, \qquad \lambda_\eta \geq 0, \sum_\eta \lambda_\eta = 1,$$

which generically has simple spectrum and shares the same eigenvectors. The ordered real Schur decomposition

$$N \;=\; QTQ^\top, \qquad Q = [q_1 \;\cdots\; q_r] \tag{33}$$

returns orthonormal vectors $q_k$ spanning the eigenvectors $w(\mu^{(k)})$. The coordinates of each atom are then read by Rayleigh quotients:

$$\mu_\eta^{(k)} \;=\; q_k^\top N_\eta q_k, \qquad \eta = 1, \ldots, m, \;\; k = 1, \ldots, r. \tag{34}$$

The above extraction procedure is summarized in Algorithm 1.

Since both $\mathcal{S}$ and $\mathcal{S}_{\mathrm{vr}}$ are nonempty, both Equation (24) and Equation (27) are always feasible (take, e.g., the Dirac measure at any $\mu \in \mathcal{S}$ or $\mathcal{S}_{\mathrm{vr}}$). We can thus solve them using the standard procedure:

---

**Algorithm 1** Extraction for single-player IREFGs

---

**Input:** Optimal solution $y^*$; order $s$ with $\operatorname{rank} M_s(y^*) = \operatorname{rank} M_{s-1}(y^*) = r$.
**Output:** $r$ optimal strategies $\{\mu^{(k)}\}_{k=1}^r$.
**1.** Factor $M_s(y^*) = VV^\top$ as in Equation (30).
**2.** Build $\mathcal{R}$ such that $v_s = \mathcal{R}w$ as in Equation (31).
**3.** For each $\mu_\eta$, construct $N_\eta$ satisfying $N_\eta\, w(\mu) = \mu_\eta\, w(\mu)$ as in Equation (32).
**4.** Form $N = \sum_\eta \lambda_\eta N_\eta$ with random $\lambda_\eta$, and compute $N = QTQ^\top$ as in Equation (33).
**5.** For each $k$, set $\mu_\eta^{(k)} = q_k^\top N_\eta q_k$ for all $\eta$ as in Equation (34).

---

**Moment-SOS loop.**   Recall $d_0 = d_0^{\mathrm{vr}} = d_u$. Initialize $d := d_0$ and do:

1. Solve the order-$d$ moment SDP; obtain optimal $y_d$ and upper bound $u_d^{\mathrm{mom}} = L_{y_d}(u)$.

2. For $s = d_0, \ldots, d$, test the rank condition in Equation (28). If it holds for some $s$, *terminate*: the relaxation is exact and one can extract the global maximizers from $M_s(y_d)$ using Algorithm 1.

3. Otherwise, increase the relaxation order: $d \leftarrow d + 1$ and go back to Step 1.

For single-player NAM-IREFGs, Lemma H.1 shows that $\operatorname{rank} M_s(y) = \operatorname{rank} M_\ell(y)\ \forall s > \ell$, and Statement (iii) of Theorem 4.1 gives exactness at degree $\ell+1$: $u_{\ell+1}^{\mathrm{mom,vr}} = u^*$. Hence the flatness test in Step 2 necessarily succeeds at $s = d = \ell+1$ (indeed for all $s > \ell$). The loop is therefore guaranteed to terminate at this fixed order, determined solely by the number of infosets, and the extraction returns at least one optimal pure strategy.

For single-player IREFGs with absentmindedness, the same loop produces a monotone sequence of upper bounds $u_d^{\mathrm{mom}} \downarrow u^*$ and terminates as soon as flatness is detected at some order $s$, which certifies exactness and enables extraction. In the absence of flatness, one increases $d$ to tighten the bound, with asymptotic convergence to $u^*$ guaranteed. If $u$ is generic, Statement (ii) of Theorem 4.1 ensures finite termination of the loop, with extraction of at least one certified global maximizer for the KKT-based problem.

## G. The Select-Verify-Cut Procedure

Recall the joint KKT system

$$\begin{cases} w_i^j(\mu) + \nu_i^j(\mu)\mathbf{1} + \lambda_i^j(\mu) = 0, \quad \lambda_i^j(\mu) \geq 0, \\ g_i^j(\mu) \geq 0, \quad h_i^j(\mu) = 0, \quad g_{i,a}^j(\mu)\,\lambda_{i,a}^j(\mu) = 0, \end{cases} \quad \forall i \in [\![n]\!],\ j \in [\![\ell_i]\!],\ a \in [\![m_i^j]\!]. \tag{35}$$

The following exposition is based on the method introduced in (Nie & Tang, 2024).

**(i) Select.**   Let $n_0 := \sum_{i=1}^n \sum_{j=1}^{\ell_i} m_i^j$ be the total dimension, set $[\mu]_1 = (1, \mu^\top)^\top$, and choose a generic positive definite matrix $\Theta \in \mathbb{R}^{(n_0+1)\times(n_0+1)}$. Then all NEs are feasible points of

$$\min_\mu\ \varphi_\Theta(\mu) := [\mu]_1^\top \Theta [\mu]_1 \quad \text{s.t.} \quad \begin{cases} \mu \text{ satisfies (KKT)}, \\ u_i(\mu_i, \mu_{-i}) - u_i(v_i, \mu_{-i}) \geq 0, \quad \forall v_i \in K_i,\ \forall i, \end{cases} \tag{36}$$

where $K_i$ is the current (finite) set of deviation profiles used as cuts (initially $K_i = \emptyset$). If Equation (36) is infeasible, there is no NE. If it is feasible, a minimizer exists because the feasible set is compact and $\varphi_\Theta$ is continuous.

**(ii) Verify.**   Let $\hat{\mu} \in \mathbb{R}^{n_0}$ be an optimizer of Equation (36). For each player $i$, evaluate the best-response improvement against $\hat{\mu}_{-i}$ by solving the KKT-restricted POP (same value as the unrestricted best-response since LICQ holds on products of simplices):

$$\begin{aligned} \omega_i := \max_{\mu_i}\ & u_i(\mu_i, \hat{\mu}_{-i}) - u_i(\hat{\mu}_i, \hat{\mu}_{-i}) \\ \text{s.t.}\ & w_i^j(\mu) + \nu_i^j(\mu)\mathbf{1} + \lambda_i^j(\mu) = 0, \quad \lambda_i^j(\mu) \geq 0, \qquad \forall j, \\ & g_i^j(\mu) \geq 0, \quad h_i^j(\mu) = 0, \quad g_{i,a}^j(\mu)\,\lambda_{i,a}^j(\mu) = 0, \qquad \forall j, a. \end{aligned} \tag{37}$$

If every $\omega_i \leq 0$, no player can profitably deviate and $\hat{\mu}$ is an NE.

**(iii) Cut.** If some $\omega_i > 0$, take one or more maximizers $v_i \in \arg\max u_i(\mu_i, \hat{\mu}_{-i}) - u_i(\hat{\mu}_i, \hat{\mu}_{-i})$ and add the valid NE cuts

$$u_i(\mu_i, \mu_{-i}) - u_i(v_i, \mu_{-i}) \geq 0 \qquad (v_i \in K_i \leftarrow K_i \cup \{v_i\}), \tag{38}$$

which every NE satisfies but $\hat{\mu}$ violates; then resolve Equation (36) with the enlarged cut set. Each violated cut eliminates the current candidate while preserving the entire NE set. Repeat (select-verify-cut) until an NE is certified or nonexistence is proved.

# H. Omitted Proofs from Main Text

## H.1. Proofs from Section 3

**Proposition 3.2.** *In non-absentminded IREFGs (NAM-IREFGs), each player's utility $u_i(\mu)$ is multi-affine in the blocks $\{\mu_i^j = (\mu_{i,a}^j)_{a=1}^{m^j}\}_{j=1}^{\ell_i}$, i.e., for any player $i$ and infoset $j$, the map $\mu_i^j \mapsto u_i(\mu)$ is affine when all other blocks $\{\mu_{i'}^{j'}\}_{(i',j')\neq(i,j)}$ are held fixed.*

*Proof.* Fix a terminal history $z \in Z$. For each player $i$, let $\mathcal{I}_i(z) \subseteq [\ell_i]$ be the set of (distinct) infosets of player $i$ visited on the unique path to $z$. By non-absentmindedness, each $I_i^j$ is visited at most once. Let $a_i^j(z)$ be the action taken at $I_i^j \in \mathcal{I}_i(z)$, and let $c(z)$ denote the product of chance move probabilities (independent of $\mu$). The reach probability factorizes as

$$\mathbb{P}(z \mid \mu) = c(z) \prod_{i'=1}^{n} \prod_{j \in \mathcal{I}_{i'}(z)} \mu_{i', a_{i'}^j(z)}^j.$$

Hence player $i$'s expected payoff is

$$u_i(\mu) = \sum_{z \in Z} \rho_i(z) \, \mathbb{P}(z \mid \mu) = \sum_{z \in Z} (\rho_i(z) c(z)) \prod_{i'=1}^{n} \prod_{j \in \mathcal{I}_{i'}(z)} \mu_{i', a_{i'}^j(z)}^j.$$

In each summand, the dependence on the block $\mu_i^j$ is either absent (if $j \notin \mathcal{I}_i(z)$) or linear through a single coordinate $\mu_{i, a_i^j(z)}^j$ (if $j \in \mathcal{I}_i(z)$); by non-absentmindedness, no monomial contains two coordinates from the same block. Therefore, with all other blocks $\{\mu_{i'}^{j'}\}_{(i',j')\neq(i,j)}$ held fixed, the map $\mu_i^j \mapsto u_i(\mu)$ is affine on the simplex $\Delta^{m_i^j}$. Since this holds for every $(i,j)$, $u_i$ is multi-affine in the blocks $\{\mu_i^j\}_{j=1}^{\ell_i}$. $\qquad\square$

## H.2. Proofs from Section 4

**Theorem 4.1.** *Consider a single-player IREFG $\mathscr{G}$ with utility function $u$. Let $\ell$ be the number of infosets, $d_0 := \max_{j,a}\{\lceil\deg(u)/2\rceil, \lceil\deg(g_a^j)/2\rceil, \lceil\deg(h^j)/2\rceil\}$, and $u^*$ be the ex-ante optimal value of $\mathscr{G}$. Denote by $u_d^{\mathrm{sos}}, u_d^{\mathrm{sos,kkt}}, u_d^{\mathrm{sos,vr}}$ the values obtained from the Moment-SOS hierarchies applied respectively to the vanilla product-of-simplices, KKT-based, and vertex-restricted formulations. Similarly, we use the superscript $\mathrm{mom}$ to denote the moment hierarchy. Then we have the following:*

*(i) $\lim_{d\to\infty} u_d^{\mathrm{sos}} = \lim_{d\to\infty} u_d^{\mathrm{mom}} = u^*$.*

*(ii) If $u$ is generic, there exists $d \geq d_0$ with $u_d^{\mathrm{mom,kkt}} = u_d^{\mathrm{sos,kkt}} = u^*$.*

*(iii) If $\mathscr{G}$ is non-absentminded, the degree-$(\ell+1)$ moment relaxation of the vertex-restricted problem is exact: $u_{\ell+1}^{\mathrm{mom,vr}} = u^*$.*

*Proof.* To prove Statement (i), note that because $Q(\mathcal{S})$ is Archimedean, the asymptotic convergence follows from Putinar's Positivstellensatz and Lasserre's hierarchy (Putinar, 1993; Lasserre, 2001; 2024).

To prove Statement (ii), recall from Section 3 that if $u$ is generic (a property which holds for almost all single-player IREFGs), the (KKT) set is finite. Augmenting Equation (1) with the polynomial (KKT) system does not change the set

of maximizers, but restricts feasibility to a finite real variety. For such finite varieties, the Lasserre hierarchy has finite convergence: for some $d$ large enough, $u_d^{\mathrm{mom,kkt}} = u_d^{\mathrm{sos,kkt}} = u^*$ and the flat extension (rank) condition holds, allowing recovery of $\mu^*$; this is immediate from e.g. (Laurent, 2008, Thm. 6.15) and (Lasserre et al., 2008, Prop. 4.6).

In order to prove Statement (iii), we first establish a key rank-stabilization lemma:

**Lemma H.1.** *For every feasible solution $y$ of the degree-$s$ moment relaxation in Equation* (27) *with $s > \ell$, it holds that* $\operatorname{rank} M_s(y) = \operatorname{rank} M_\ell(y)$.

*Proof of Lemma H.1.* Let $v_s(\mu)$ collect all monomials of total degree $\leq s$ and recall $M_s(y) = L_y(v_s v_s^\top)$. Index the columns of $M_s(y)$ by monomials and write the block decomposition

$$M_s(y) = \begin{bmatrix} M_\ell(y) & B \\ B^\top & C \end{bmatrix},$$

where $M_\ell(y)$ is indexed by monomials of degree $\leq \ell$ and $B$ by monomials of degree $> \ell$.

Fix a column of $B$ indexed by $m(\mu) = \prod_{j=1}^\ell \prod_{a=1}^{m^j} (\mu_a^j)^{\alpha_a^j}$ with $\sum_{j,a} \alpha_a^j = \deg m > \ell$. Define the clipped monomial

$$\widehat{m}(\mu) := \prod_{j=1}^\ell \prod_{a=1}^{m^j} (\mu_a^j)^{\min\{1,\alpha_a^j\}},$$

so each exponent $\geq 1$ is replaced by 1.

By repeatedly using $L_y\big((\mu_a^j)^2 q\big) = L_y(\mu_a^j q)$ (i.e., $b_a^j = 0$), for any row index monomial $r$ (degree $\leq s$) we obtain

$$L_y\big(m(\mu)\, r(\mu)\big) = L_y\big(\widehat{m}(\mu)\, r(\mu)\big).$$

Hence the column of $M_s(y)$ indexed by $m$ coincides with the column indexed by $\widehat{m}$.

If $\widehat{m}$ uses at most one variable per block, then $\deg \widehat{m} \leq \ell$ and the column indexed by $m$ is *identical* to a column of $M_\ell(y)$. If, instead, $\widehat{m}$ contains two distinct variables from the same block (say $\mu_a^j$ and $\mu_{a'}^j$ with $a \neq a'$), then $\widehat{m}$ vanishes on the vertex set $\mathcal{S}_{\mathrm{vr}}$ (one-hot per block), so for all admissible rows $r$, $L_y(\widehat{m}\, r) = 0$, and the entire column indexed by $m$ is the zero vector.

Consequently, every column of $B$ is either zero or identical to a column of $M_\ell(y)$. Applying the same argument to the lower block, with $A := \begin{bmatrix} M_\ell(y) \\ B^\top \end{bmatrix}$ and $D := \begin{bmatrix} B \\ C \end{bmatrix}$, shows that every column of $D$ is either zero or identical to a column of $A$. Therefore the column space of $M_s(y)$ is contained in the column space of $A$, hence $\operatorname{rank} M_s(y) \leq \operatorname{rank} A = \operatorname{rank} M_\ell(y)$. The reverse inequality is obvious because $M_\ell(y)$ is a principal submatrix of $M_s(y)$. Hence $\operatorname{rank} M_s(y) = \operatorname{rank} M_\ell(y)$ for all $s > \ell$. $\qquad\square$

Let $s := \ell+1$ and let $y^*$ be an optimal solution of the order-$s$ moment relaxation Equation (27). By Lemma H.1, $\operatorname{rank} M_s(y^*) = \operatorname{rank} M_\ell(y^*)$, so the flatness condition holds at order $s$.

Set $r := \operatorname{rank} M_s(y^*)$. By (Curto & Fialkow, 2000, Theorem 1.6), $y^*$ admits an $r$-atomic representing measure $\sum_{k=1}^r \lambda_k \delta_{\mu^{(k)}}$ supported on the feasible set, with $\lambda_k > 0$ and $\sum_k \lambda_k = 1$. Moreover, the equalities $L_{y^*}(h^j q) = L_{y^*}(b_a^j q) = 0$ in Equation (27) enforce that the atoms of the representing measure lie in $S_{\mathrm{vr}}$. Therefore,

$$L_{y^*}(u) = \sum_{k=1}^r \lambda_k\, u(\mu^{(k)}) \leq \max_{\mu \in \mathcal{S}_{\mathrm{vr}}} u(\mu) = u^*.$$

Because Equation (27) is a relaxation of the original problem, $u_s^{\mathrm{mom,vr}} \geq u^*$ for all $s$. At $s = \ell+1$, we have

$$u^* \leq u_{\ell+1}^{\mathrm{mom,vr}} = L_{y^*}(u) \leq u^*.$$

Hence $u_{\ell+1}^{\mathrm{mom,vr}} = u^*$ and the optimum is attained at $y^*$. $\qquad\square$

## H.3. Proofs from Section 5

**Theorem 5.1.** *Let $\mathcal{G}$ be a multi-player IREFG with utility functions $u_i$ for each player $i$. Throughout, subproblems are solved by the KKT-based hierarchies of increasing order. Then, we have the following:*

*(i) The SVC procedure is asymptotically exact: as the relaxation order and number of iterations grow, it returns a behavioral NE when one exists, and otherwise a certificate of nonexistence.*

*(ii) If $u_i$ are all generic, the KKT-based hierarchy has finite convergence for all SVC subproblems, and the SVC loop terminates in finitely many iterations.*

*(iii) If $\mathcal{G}$ is non-absentminded, the Verify/Cut phases in SVC are unnecessary: a single vertex-restricted Select (Equation (7)) suffices to compute an NE or certify nonexistence. Its Moment-SOS hierarchy is asymptotically exact; if $u_i$ are generic, it attains exactness at a finite order.*

*Proof.* (i) Let $\ell_0 := \sum_{i=1}^{n} \ell_i$ be the total number of infosets. For any feasible $\mu$ we have, for each infoset block, $\sum_{a=1}^{m_i^j} \mu_{i,a}^j = 1$ and $0 \leq \mu_{i,a}^j \leq 1$. Hence

$$\|\mu\|^2 = \sum_{i=1}^{n} \sum_{j=1}^{\ell_i} \sum_{a=1}^{m_i^j} (\mu_{i,a}^j)^2 \;\leq\; \sum_{i=1}^{n} \sum_{j=1}^{\ell_i} \sum_{a=1}^{m_i^j} \mu_{i,a}^j \;=\; \sum_{i=1}^{n} \sum_{j=1}^{\ell_i} 1 \;=\; \ell_0,$$

so $g_0(\mu) := \ell_0 - \|\mu\|^2 \geq 0$ on $\mathcal{S}$. Adding $g_0 \geq 0$ yields an Archimedean quadratic module, and the same holds for each verification feasible set. By standard results for Lasserre's hierarchy on Archimedean sets (see, e.g., (Lasserre, 2001; Laurent, 2008)), every fixed selector (with a fixed cut set) and every verification problem is asymptotically exact: the moment optimal values converge to the true optima as $d \to \infty$, flat truncation recovers optimizers, and infeasibility is detected at high order.

At loop $t$, solve one selector and up to $n$ verifications. If the selector becomes infeasible at some order, nonexistence is certified and the procedure stops. Otherwise, let $\hat{\mu}^{(t)}$ be a selector optimizer recovered once flatness occurs. If all verification values are $\leq 0$, then $\hat{\mu}^{(t)}$ is an NE and we stop. If some player gains ($> 0$), extract one or more violated valid inequalities $u_i(\mu_i, \mu_{-i}) - u_i(v_i, \mu_{-i}) \geq 0$ from the deviator $v_i$ and add them to the selector. As relaxation orders increase across loops, subproblem solutions approach their true optima; any limit point of flat selector solutions satisfies all accumulated valid inequalities, i.e., the Nash conditions. Hence the method converges asymptotically to an NE, or certifies nonexistence if the selector turns infeasible.

(ii) Under generic utilities, which hold for almost all IREFGs, the joint KKT set is finite (cf. (Nie & Tang, 2024)). Then the selector's feasible set (joint KKT plus cuts) is finite. Each failed candidate is removed by the new valid inequalities without excluding any NE, so only finitely many candidates can be visited before selecting an NE or proving infeasibility. On finite feasible sets, the Moment-SOS hierarchy attains finite convergence and yields atomic solutions (see, e.g., (Laurent, 2008; Lasserre et al., 2008)). Therefore both the select-verify-cut loop and its SDP subproblems terminate in finitely many steps.

(iii) In the NAM case, $u_i(\cdot, \mu_{-i})$ is linear in each block $\mu_i^j$ (Proposition 3.2), so "no profitable deviation by $i$" $\iff$ $u_i(\mu_i, \mu_{-i}) \geq u_i(v_i, \mu_{-i}) \; \forall v_i \in \mathcal{S}_{i,\mathrm{vr}}$. Thus feasibility of the single vertex-restricted selector Equation (7) is equivalent to the existence of a behavioral NE, and the verify/cut phases are unnecessary.

To see asymptotic exactness, note that the feasible region of Equation (7) is contained in the product of simplices (per-player KKT equalities are imposed together with vertex deviation inequalities). Similar to (i), adding $g_0(\mu) = \ell_0 - \|\mu\|^2 \geq 0$ makes the quadratic module Archimedean. By standard results for Lasserre's hierarchy on Archimedean sets, the Moment-SOS relaxation of Equation (7) is asymptotically exact; flatness yields extraction, and infeasibility is detected at sufficiently high order.

If, moreover, the utilities $u_i$ are generic, the joint KKT variety over the product of simplices is finite. Since Equation (7) enforces these KKT equalities and further filters candidates by the vertex deviation inequalities, its feasible set is a finite real variety. On finite varieties, the Moment-SOS hierarchy attains exactness at some finite order, hence Equation (7) has finite convergence (returning an NE when feasible, and otherwise certifying nonexistence). □

## H.4. Proofs from Section 6

**Single-player IREFGs.** First, we note that single-player IREFGs can be viewed as continuous identical-interest games (see e.g. (Von Stengel & Koller, 1997)), so the existence of ex-ante optima does not require concavity: $\mathcal{S}$ is compact and $u$

is continuous, hence $\arg\max_{\mathcal{S}} u \neq \emptyset$. We establish that in this setting, there is an equivalence between the definitions of concave and monotone games.

**Proposition H.2.** *A single-player IREFG $\mathscr{G}$ is monotone if and only if the expected utility $u$ is concave on $\mathcal{S}$. Moreover, $\mathscr{G}$ is strictly monotone if and only if $u$ is strictly concave on $\mathcal{S}$.*

*Proof.* (Concavity $\Rightarrow$ Monotonicity). Assume $u$ is concave. The first-order concavity inequality gives, for all $\mu, \nu$,

$$u(\mu) \ \leq \ u(\nu) + \nabla u(\nu)^\top (\mu - \nu), \qquad u(\nu) \ \leq \ u(\mu) + \nabla u(\mu)^\top (\nu - \mu).$$

Adding the two inequalities yields

$$(\nabla u(\mu) - \nabla u(\nu))^\top (\mu - \nu) \ \leq \ 0,$$

which is exactly the definition of monotonicity with $v = \nabla u$.

(Monotonicity $\Rightarrow$ Concavity). Assume $\langle v(\mu) - v(\nu), \mu - \nu \rangle \leq 0$, $\forall \mu, \nu \in \mathcal{S}$ (i.e. the pseudogradient is monotone). Fix $\mu, \nu \in \mathcal{S}$ and set $\gamma(t) = \nu + t(\mu - \nu)$ for $t \in [0, 1]$. Define $g(t) := u(\gamma(t))$. Then $g'(t) = \nabla u(\gamma(t))^\top (\mu - \nu)$. For $0 \leq s < t \leq 1$,

$$g'(t) - g'(s) = \big(\nabla u(\gamma(t)) - \nabla u(\gamma(s))\big)^\top (\mu - \nu) = \big\langle v(\gamma(t)) - v(\gamma(s)), \gamma(t) - \gamma(s) \big\rangle \leq 0,$$

so $g'$ is nonincreasing on $[0, 1]$. Therefore

$$u(\mu) - u(\nu) = \int_0^1 g'(t)\, dt \ \leq \ \int_0^1 g'(0)\, dt = \nabla u(\nu)^\top (\mu - \nu),$$

which is the first-order characterization of concavity, hence $u$ is concave on $\mathcal{S}$.

(Strict case). If equality in $\langle v(\mu) - v(\nu), \mu - \nu \rangle \leq 0$, $\forall \mu, \nu \in \mathcal{S}$ holds only for $\mu = \nu$, then for any $\mu \neq \nu$ and any $t \in (0, 1]$ we have

$$g'(t) - g'(0) = \big\langle v(\gamma(t)) - v(\gamma(0)), \gamma(t) - \gamma(0) \big\rangle < 0,$$

so $g'$ is strictly decreasing and $u(\mu) - u(\nu) = \int_0^1 g'(t)\, dt < \nabla u(\nu)^\top (\mu - \nu)$. This is the strict first-order concavity inequality, hence $u$ is strictly concave. The converse (strict concavity $\Rightarrow$ strict monotonicity for $\mu \neq \nu$) follows by repeating the first part with inequalities strict. $\square$

**Theorem 6.3.** *Consider a single-player IREFG $\mathscr{G}$ with utility function $u$. Let $d_0 := \max_{j,a}\{\lceil \deg(u)/2 \rceil, \lceil \deg(g_a^j)/2 \rceil, \lceil \deg(h^j)/2 \rceil\}$. Then, the following holds:*

  *(i) If $\mathscr{G}$ is strictly concave/monotone, then the Moment-SOS hierarchy has finite convergence: there exists $d \geq d_0$ such that $u_d^{\mathrm{sos}} = u_d^{\mathrm{mom}} = u^*$.*

 *(ii) If $\mathscr{G}$ is SOS-concave/SOS-monotone, the degree-$d_0$ Moment-SOS relaxations are exact: $u_{d_0}^{\mathrm{mom}} = u_{d_0}^{\mathrm{sos}} = u^*$, i.e., the Moment-SOS hierarchy converges at the first level.*

*Proof.* (i) Let $\mu^* \in \mathcal{S}$ be a global maximizer, so $u^* = u(\mu^*)$. Since $\mathcal{S}$ is a nonempty polyhedron and by Proposition H.2 $u$ is (strictly) concave, there exist KKT multipliers $\{\lambda_a^j\}_{j,a}$ with $\lambda_a^j \geq 0$ and $\{\nu^j\}_j$ such that

$$\nabla u(\mu^*) + \sum_{j,a} \lambda_a^j \, \nabla g_a^j(\mu^*) + \sum_j \nu^j \, \nabla h^j(\mu^*) = 0, \quad \lambda_a^j \, g_a^j(\mu^*) = 0.$$

Let $I_m$ be the $m \times m$ identity. Since $-\nabla^2 u \succ 0$ on $\mathcal{S}$, the (strictly positive) smallest eigenvalue of $-\nabla^2 u(\mu)$ is continuous in $\mu$, and the compactness of $\mathcal{S}$ implies that there exists $\delta > 0$ such that $-\nabla^2 u(\mu) \ \succeq \ \delta I_m$ for all $\mu \in \mathcal{S}$. Define the (convex) Lagrangian-type polynomial

$$G(\mu) \ := \ u(\mu^*) - u(\mu) - \sum_{j,a} \lambda_a^j \, g_a^j(\mu) - \sum_j \nu^j \, h^j(\mu).$$

Then $G(\mu^*) = 0$, $\nabla G(\mu^*) = 0$. Define

$$F(\mu, \mu^*) \ := \ \int_0^1 \left( \int_0^t \nabla^2 G\big(\mu^* + s(\mu - \mu^*)\big)\, ds \right) dt,$$

so that the identity holds (Helton & Nie, 2010):

$$G(\mu) = G(\mu^*) + \nabla G(\mu^*)(\mu - \mu^*) + (\mu - \mu^*)^\top F(\mu, \mu^*)(\mu - \mu^*)$$
$$= \langle \mu - \mu^*, \, F(\mu, \mu^*) \, (\mu - \mu^*) \rangle.$$

Since $\nabla^2 G(\mu) = -\nabla^2 u(\mu) \succeq \delta I_m$ on $\mathcal{S}$, for any $\xi \in \mathbb{R}^m$ we have

$$\xi^T F(\mu, \mu^*) \xi \geq \delta \int_0^1 \int_0^t \xi^T \xi \, ds dt = \frac{\delta}{2} \xi^T \xi.$$

Hence $F(\mu, \mu^*) \succeq \frac{\delta}{2} I_m$ for all $\mu \in \mathcal{S}$. Since $F(\mu, \mu^*)$ is a symmetric polynomial matrix that is positive definite on $\mathcal{S}$, the matrix polynomial version of Putinar's Positivstellensatz yields SOS-matrix polynomials $F_0$, $\{F_a^j\}$ and polynomial matrices $\{H^j\}$ such that

$$F(\mu, \mu^*) = F_0(\mu) + \sum_{j,a} F_a^j(\mu) \, g_a^j(\mu) + \sum_j H^j(\mu) \, h^j(\mu).$$

Multiply it on both sides by $(\mu - \mu^*)$ to obtain

$$G(\mu) = \sigma_0(\mu) + \sum_{j,a} \sigma_a^j(\mu) \, g_a^j(\mu) + \sum_j p^j(\mu) \, h^j(\mu),$$

where $\sigma_0(\mu) := \langle \mu - \mu^*, F_0(\mu)(\mu - \mu^*) \rangle \in \Sigma[\mu]$, $\sigma_a^j(\mu) := \langle \mu - \mu^*, F_a^j(\mu)(\mu - \mu^*) \rangle \in \Sigma[\mu]$, $p^j(\mu) := \langle \mu - \mu^*, H^j(\mu)(\mu - \mu^*) \rangle \in \mathbb{R}[\mu]$. Recalling the definition of $G$ and rearranging,

$$u(\mu^*) - u(\mu) = \sigma_0(\mu) + \sum_{j,a} \underbrace{\left(\sigma_a^j(\mu) + \lambda_a^j\right)}_{\in \Sigma[\mu]} g_a^j(\mu) + \sum_j \underbrace{\left(p^j(\mu) + \nu^j\right)}_{\in \mathbb{R}[\mu]} h^j(\mu).$$

Let $d = \max\{\lceil \deg(\sigma_0)/2 \rceil, \lceil \deg(\sigma_a^j)/2 \rceil, \lceil \deg(p^j)/2 \rceil\} + 1$. Then with $u^* = u(\mu^*)$, the tuple $(u^*, \sigma_0, \{\sigma_a^j + \lambda_a^j\}, \{p^j + \nu^j\})$ is feasible for the degree-$d$ SOS program in Equation (15), so $u_d^{\text{sos}} \leq u^*$. By weak duality, we have $u_d^{\text{sos}} \geq u_d^{\text{mom}} \geq u^*$. Therefore, $u_d^{\text{sos}} = u_d^{\text{mom}} = u^*$. Conversely, choosing $y$ as the Dirac moments of $\delta_{\mu^*}$ in the moment SDP of Equation (24) gives a feasible point with value $L_y(u) = u^*$.

(ii) Recall $d_0 = \max\{d_u, d_{\mathcal{S}}\} = d_u$. Let $\mu^* \in \arg\max_{\mu \in \mathcal{S}} u(\mu)$ and set $u^* := u(\mu^*)$. Because $\mathscr{G}$ is (SOS-)concave/monotone, Proposition H.2 implies that $u$ is concave on $\mathcal{S}$. Since $\mathcal{S}$ is a nonempty polyhedron, the KKT conditions are necessary and sufficient for optimality. Hence, for any optimal solution $\mu^*$ there exist Lagrange multipliers $\{\lambda_a^j\}_{j,a}$ with $\lambda_a^j \geq 0$ and $\{\nu^j\}_j$ such that:

$$\nabla(u)(\mu^*) + \sum_{j,a} \lambda_a^j \, \nabla g_a^j(\mu^*) + \sum_j \nu^j \, \nabla h^j(\mu^*) = 0,$$
$$\lambda_a^j \, g_a^j(\mu^*) = 0, \quad g_a^j(\mu^*) \geq 0, \quad \lambda_a^j \geq 0, \quad h^j(\mu^*) = 0.$$

Since $-\nabla^2 u$ is SOS-matrix and $\deg(-u) \leq 2d_0$, by (Lasserre, 2024, Theorem 3.9):

$$(-u)(\mu) - (-u)(\mu^*) - \nabla(-u)(\mu^*)^T(\mu - \mu^*) = \sigma_0(\mu),$$

with $\sigma_0 \in \Sigma[\mu]_{d_0}$. Using stationarity and linearity of $g_a^j, h^j$ (their gradients are constant), we obtain from the KKT condition:

$$\nabla(u)(\mu^*)^\top(\mu - \mu^*) = -\sum_{j,a} \lambda_a^j \nabla(g_a^j)^\top(\mu - \mu^*) - \sum_j \nu^j \nabla(h^j)^\top(\mu - \mu^*)$$
$$= -\sum_{j,a} \lambda_a^j(g_a^j(\mu) - g_a^j(\mu^*)) - \sum_j \nu^j h^j(\mu),$$

where we used $h^j(\mu^*) = 0$. Plugging this back, we have

$$u^* - u(\mu) = \sigma_0(\mu) + \sum_{j,a} \lambda_a^j \, (g_a^j(\mu) - g_a^j(\mu^*)) + \sum_j \nu^j \, h^j(\mu)$$
$$= \underbrace{\sigma_0(\mu)}_{\in \Sigma[\mu]} + \sum_{j,a} \underbrace{\lambda_a^j}_{\in \Sigma[\mu]} g_a^j(\mu) + \sum_j \nu^j \, h^j(\mu).$$

since $\sum_{j,a} \lambda_a^j g_a^j(\mu^*) = 0$. Thus we have the original-domain SOS certificate

$$u^* - u(\mu) \;=\; \sigma_0(\mu) + \sum_{j,a} \sigma_a^j(\mu)\, g_a^j(\mu) + \sum_j p^j(\mu)\, h^j(\mu),$$

with $\sigma_a^j(\mu) \equiv \lambda_a^j$ (nonnegative constants are SOS) and $p^j(\mu) \equiv \nu^j$. Degree bounds: $\deg(\sigma_0) \le 2d_0$, $\deg(\sigma_a^j g_a^j) \le 1$, $\deg(p^j h^j) \le 1$. Thus $(u^*, \sigma_0, \{\sigma_a^j\}, \{p^j\})$ is feasible for the SOS dual Equation (15) at order $d_0$, yielding $u_{d_0}^{\mathrm{sos}} \le u^*$. By weak duality, we have $u_{d_0}^{\mathrm{sos}} \ge u_{d_0}^{\mathrm{mom}} \ge u^*$. Therefore $u_{d_0}^{\mathrm{mom}} = u_{d_0}^{\mathrm{sos}} = u^*$. Conversely, choosing $y$ as the Dirac moments of $\delta_{\mu^*}$ in the moment SDP Equation (24) gives a feasible point with value $L_y(u) = u^*$. □

# I. Additional Empirical Examples

In this section, we show some illustrative examples for how our proposed methods can be used to compute ex-ante optima in single-player IREFGs. We remark that we use only standard scientific computing packages in Julia, alongside an off-the-shelf SumOfSquares package (Legat et al., 2017; Weisser et al., 2019). The code is run on a PC with an AMD Ryzen 5 5600 processor and 16 GB of RAM running a 64-bit version of Windows 11, and is provided in a supplementary file.

**Example C.4.** As a running example, we revisit Example C.4. Since the game is not multilinear (i.e. the player is absentminded), we use the standard Moment-SOS hierarchies. In particular, we run the SOS hierarchy in the Moment-SOS loop, testing the rank condition at each level until an atomic measure (i.e. a feasible maximizer) can be extracted. The program converges at truncation degree $D_{\mathrm{SOS}} = 4$, returning optimal solution $(x_{11}^*, x_{12}^*) = (1, 0)$ and $(x_{21}^*, x_{22}^*) = (1, 0)$. This gives objective value $p(x^*) = 9$. The compute time required to solve this example was 0.02 seconds.

**Randomly Generated NAM-IREFG.** We also create a procedure to randomly generate single-player IREFGs. Specifically, we seek to validate Statement (iii) of Theorem 4.1, that convergence occurs at a structure dependent level of the Moment-SOS hierarchy. For example, consider a (randomly generated) non-absentminded game $\mathscr{G}_1$ with 3 infosets and two actions per infoset, resulting in variables $x_1, x_2$ for $I_1$, $y_1, y_2$ for $I_2$, and $z_1, z_2$ for $I_3$. The payoff function for $\mathscr{G}_1$ is given by:

$$u_{\mathscr{G}_1}(x, y, z) = -4z_1 + x_2 y_2 + x_2 y_2 z_1 - 3x_2 y_1 z_2 - 3x_2 y_1 z_1. \tag{39}$$

Due to Theorem 4.1, we expect convergence and extraction to be possible at level $d = 4$ of the hierarchy, since there are 3 infosets. Moreover, since the game is a NAM-IREFG, we can further restrict the feasible region to vertex set $\mathcal{S}_{\mathrm{vr}}$. Setting $d = 4$ in the hierarchy, we find ex-ante optimal solution $(x_1^*, x_2^*) = (0, 1)$, $(y_1^*, y_2^*) = (0, 1)$, and $(z_1^*, z_2^*) = (0, 1)$ with optimal value 1. The compute time required to solve this game was 0.06 seconds.

**Randomly Generated Absentminded IREFG.** As another example, we show that in a randomly generated absentminded game, the hierarchies empirically converge at 'reasonable' levels. Consider game $\mathscr{G}_2$ with two infosets $I_1$ and $I_2$, where the player chooses between 3 actions in each infoset. This gives variables $x_1, x_2, x_3$ in $I_1$ and $y_1, y_2, y_3$ in $I_2$. The payoff function for $\mathscr{G}_2$ is given by

$$u_{\mathscr{G}_2}(x, y) = 4x_1 x_3 + 2x_2 x_3 y_3 - 5x_1 x_2 y_3 + x_1 x_2 y_1 - 4x_2 x_3 y_2 y_3. \tag{40}$$

Running the hierarchies, we obtain convergence at truncation degree $D_{\mathrm{SOS}} = 6$, with ex-ante optimal solution $(x_1^*, x_2^*, x_3^*) = (0.5, 0, 0.5)$ and $(y_1^*, y_2^*, y_3^*) = (0.134, 0.594, 0.272)$, giving optimal value 1. Notice that unlike the NAM case, the optimal solution is not a vertex. The total compute time required to solve this example was 0.41 seconds.

**Generalized Absentminded-Driver Benchmark.** To complement the randomly generated polynomial instances in the main text, we introduce an extended benchmark family based on the classic absentminded-driver game. This family generalizes the one-variable example to multiple infosets and multi-action decision points. The game has $\ell$ repeated infosets $I_1, \ldots, I_\ell$, where infoset $I_j$ is visited $d_j$ times along the relevant history. At each $I_j$, there is one continuation action and $m_j - 1$ terminal actions. The behavioral strategy at $I_j$ is $x_j = (x_{j,0}, x_{j,1}, \ldots, x_{j,m_j-1}) \in \Delta^{m_j-1}$, where $x_{j,0}$ denotes the probability of taking the continuation action. If terminal action $a_{j,b}$ is chosen at the $k$-th visit of $I_j$, the payoff is $r_{j,k,b}$. If the agent always continues through all infosets, it receives terminal penalty $r_{\mathrm{term}} = -1$.

The induced ex-ante utility is

$$u(x_1, \ldots, x_\ell) = \sum_{j=1}^{\ell} \left( \prod_{h<j} x_{h,0}^{d_h} \right) \left( \sum_{k=1}^{d_j} x_{j,0}^{k-1} \sum_{b=1}^{m_j-1} r_{j,k,b} x_{j,b} \right) + r_{\text{term}} \prod_{h=1}^{\ell} x_{h,0}^{d_h}.$$

This is a polynomial over $\prod_{j=1}^{\ell} \Delta^{m_j-1}$ with total degree $D = \sum_{j=1}^{\ell} d_j$. The construction is genuinely imperfect-recall and absentminded whenever some $d_j > 1$, since the same infoset is revisited multiple times along a single history.

As a representative benchmark, we set $\ell = 3$, $d = (2, 2, 2)$, and $m = (2, 3, 4)$, yielding a degree-6 polynomial instance with 9 behavioral variables. We generated 100 random instances by sampling the payoffs $r_{j,k,b}$ uniformly from $[-1, 1]$ and fixing $r_{\text{term}} = -1$. Using SOS truncation degree $D_{\text{SOS}} = 6$, equivalently Moment-SOS relaxation order $d = 3$, we extracted atomic, optimal solutions in all $100/100$ instances, with an average runtime of 1.56s.

**SOS-Monotone Example.** We show experimental corroboration for Statement (ii) of Theorem 6.3. Using a technique established in (Ahmadi et al., 2013), we construct a game $\mathscr{G}_{\text{SOS}}$ with degree-4 polynomial utility which is SOS-concave. The polynomial is given below:

$$\begin{aligned}
u_{\mathscr{G}_{\text{SOS}}}(x, y) = {} & 9.37y_2^4 + 9.37y_1^2 y_2^2 + 9.37y_1^4 + 1.17x_2^2 y_2^2 - 0.09x_2^2 y_1 y_2 + 0.94x_2^2 y_1^2 \\
& + 9.37x_2^4 - 0.78x_1 x_2 y_2^2 - 0.52x_1 x_2 y_1 y_2 + 0.55x_1 x_2 y_1^2 + 0.13x_1^2 y_2^2 \\
& + 0.16x_1^2 y_1 y_2 + 0.13x_1^2 y_1^2 + 9.37x_1^2 x_2^2 + 9.37x_1^4
\end{aligned} \tag{41}$$

Even though this polynomial is quartic, we need only run the SOS hierarchy at truncation degree $D_{\text{SOS}} = 4$, equivalently Moment-SOS relaxation order $d = 2$, to obtain the optimal value and extract a solution. We obtain the solution $(x_1^*, x_2^*) = (0, 1)$ and $(y_1^*, y_2^*) = (0, 1)$, with value 19.9. The compute time was $< 0.001$ seconds.

