# OpenReview forum: "Solving Imperfect-Recall Games via Sum-of-Squares Optimization"
_ICML.cc/2026/Conference — ICML 2026 regular_

### Official Review · Reviewer_XaMZ · 2026-03-04

**Soundness:** 2
**Presentation:** 2
**Significance:** 3
**Originality:** 3
**Overall Recommendation:** 4
**Confidence:** 3

**Summary:**

This paper explores the computationally hard problem of finding equilibria in imperfect-recall extensive-form games (IREFGs). By rigorously formulating IREFGs as polynomial optimization problems, the authors introduce the Moment-Sum-of-Squares (SOS) hierarchy as a core solution framework. The paper provides strong theoretical guarantees, proving asymptotic convergence for general games and exact finite convergence for generic IREFGs. For non-absentminded (NAM) games, the authors prove that exact convergence occurs at a finite degree strictly bounded by the number of information sets. Furthermore, the authors propose a novel Select-Verify-Cut (SVC) algorithmic framework for multi-player equilibrium computation and identify tractable game subclasses (SOS-monotone/concave) solvable at the very first level of the hierarchy.

**Compliance With Llm Reviewing Policy:**

Affirmed.

**Final Justification:**

I update my weak reject score to weak accept. The paper makes a meaningful theoretical contribution through a rigorous SOS-based framework and strong convergence guarantees for imperfect-recall extensive-form games, and the authors’ latest clarifications increase my confidence in the scope of these results. I still have concerns about the presentation clarity and the limited empirical support, but I now view these as weaknesses that limit impact rather than reasons to keep the paper below the bar.

**Key Questions For Authors:**

Questions:

Q1. Could you explicitly define the genericity conditions in the main text? Are there any theoretical insights or practical heuristics to handle these degenerate cases?

Q2. Are there any theoretical guarantees, or at least empirical observations, regarding the approximation error of the extracted strategies at these lower, tractable levels?

Q3. Can you provide an evaluation of your method on at least one specific game example rather than a manually engineered polynomial?

**Limitations:**

Not sufficiently. Authors should state the assumption and discuss computational limitation more clearly, see weaknesses.

**Strengths And Weaknesses:**

Strengths:

S1. The paper addresses the computation of Nash equilibria in IREFGs, a notoriously challenging problem that is known to be NP-hard. Instead of resorting to purely heuristic or approximate methods, the authors provide a systematic computational framework with rigorous mathematical guarantees for this highly complex domain.

S2. The authors elegantly bridge game theory and algebraic geometry. They successfully deploy the Moment-Sum-of-Squares (SOS) hierarchy to the strategic setting of extensive-form games.

S3. The observation that Non-Absentminded IREFGs inherently yield multi-affine utility polynomials is valuable. The authors cleverly utilize this property to prove a strict, exact finite convergence bound at level $d = l+1$, which is a significant theoretical advancement over the standard. The paper also identifies tractable subclasses: SOS-concave and SOS-monotone IREFGs, and proves that these subclasses can be solved exactly at the very first hierarchy level, which is interesting and beautiful result.

Weaknesses:

W1. The paper heavily relies on a genericity assumption to guarantee finite convergence. While the authors are mathematically correct that degenerate cases form a measure-zero set in the space of all polynomials, this argument seems restrictive. Several settings inherently feature integer payoffs, pervasive ties, and symmetric structures. These properties frequently induce continuous sets of equilibria (infinite KKT points), landing exactly on the "measure-zero" algebraic variety. Furthermore, more importantly, the precise mathematical definition of this genericity assumption is somewhat vaguely presented in the main text, making it hard for readers to assess its restrictiveness.

W2. The core theoretical contribution appears to be formalizing IREFGs into Polynomial Optimization and applying existing SOS/moment hierarchies. While theoretically sound, the resulting algorithm is inherently intractable for general cases because the SDP size scales exponentially. Although the authors discuss certain tractable subsets (e.g., SOS-concave/monotone), the general value of the algorithm remains questionable. An analysis of the approximation quality when the hierarchy is truncated at a computationally feasible, low level before exact convergence is reached will be helpful. Without understanding the behavior and approximation bounds of these low-level relaxations, the practical value of this framework is difficult to justify.

W3. The experimental section is overly simplistic and insufficient to demonstrate the method's efficacy. First, the evaluation is performed on manually constructed mathematical polynomial utilities rather than genuine game-theoretic topologies (e.g., the classic absentminded driver game, or an imperfect-recall poker abstraction). This harms readability a lot. Second, the experiments only test extremely trivial scales and effectively avoid the scalability wall of SOS optimization. Although the authors briefly touch upon scalability in the discussion, the limitations are not transparently thoroughly investigated. A proper empirical stress test showing the exact boundary of tractability (where the method runs out of time or memory) is missing.

---

> ### Author Rebuttal · Authors · 2026-03-31
>
> We are grateful to the reviewer for the thoughtful comments and questions. We will respond to each of your concerns separately below.
> > W1/Q1: On genericity assumption and dealing with degeneracy.
>
> We agree that the genericity assumption should be stated more explicitly. The definition of genericity in terms of a Lebesgue measure-zero set is standard, see, e.g., [1], and is intended to characterize a property that holds almost everywhere. A more concrete and practically useful sufficient condition is that the property fails only on a proper algebraic variety, that is, on the solution set of a system of polynomial equations. Indeed, every proper algebraic variety has Lebesgue measure zero [2], and therefore any property that fails only on such a set is generic. For example, the property that a square matrix is invertible is generic, since it fails precisely when its determinant vanishes, and the set of singular matrices is the proper algebraic variety defined by the polynomial equation $\det(A)=0$.
>
> In our setting, the notion of a generic game ensures that the players’ best-response problems have only finitely many KKT solutions, after fixing the game structure and the admissible utility class. For instance, our framework applies to *generic* zero-sum games. Concretely, this means that the first $n-1$ utility functions are chosen outside a measure-zero set (i.e., outside a proper algebraic variety), while the last utility function is then determined by the condition that the utilities sum to zero. We will add a clearer definition and example in subsequent versions of the paper.
>
> A possible method to deal with degenerate cases is to solve a nearby perturbed game $u_i^\varepsilon=u_i+\varepsilon p_i$, where $p_i$ is a polynomial with randomly chosen coefficients. Since the nongeneric set is algebraic of measure zero, such perturbations are generic almost surely and often isolate the KKT candidate set. We emphasize, however, that this is only a heuristic: it solves a nearby perturbed game rather than the original one, so it is not part of our formal theorem.
>
> *[1] J. Nie and K. Ranestad. Algebraic degree of polynomial optimization. SIAM J. Optim., 2009.*
>
> *[2] J. Harris. Algebraic Geometry: A First Course. Springer, 2013.*
>
> > W2/Q2: Approximation error at low-level relaxations.
>
> There is substantial work on the approximation error as a function of the hierarchy level. Here $r$ is the relaxation order, and known Putinar-type results give error decay of the form $O(1/r^c)$, with $c>0$ depending on the problem class. See Table 2 in the recent survey [3] for a concise overview.
>
> *[3] M. Laurent and L. Slot. An overview of convergence rates for sum of squares hierarchies in polynomial optimization. ICIAM, 2023.*
>
> > W3/Q3: Evaluation on specific game examples.
>
> Our original experiments were designed to test the global convergence behavior of the proposed method on randomly generated instances rather than on canonical IREFG topologies. However, this still remains within the IREFG setting: by Thm. C.3, polynomial utilities and IREFGs are equivalent, so solving the induced polynomial problem still amounts to solving an imperfect-recall game.
>
> Nevertheless, to make this link more explicit, we will add an imperfect-recall example based on a 10-stage absentminded-driver game with a single repeated information set and binary actions (Exit/Continue). If $p\in[0,1]$ denotes the probability of playing Exit, and $(r_1,\ldots,r_{10},r_{11})$ are the stage payoffs, then the induced ex-ante utility is
> $$
> u(p)=\sum_{k=1}^{10}r_k p(1-p)^{k-1}+r_{11}(1-p)^{10},
> $$
> a degree-10 polynomial induced directly by the game tree. On 100 randomly generated instances from this family, our degree-10 SOS relaxation extracted an atomic solution in all 100/100 cases, with average runtime $0.047$s. We will include this experiment in the revision as an example of solving a more challenging version of a canonical IREFG.
>
> > W3: Empirical stress test.
>
> We wish to clarify that our goal is to obtain globally Nash equilibria in a notoriously NP-hard class of games. Our primary contribution is theoretical, and the experiments are intended mainly to support the claimed global-optimality guarantees. We nevertheless agree that larger-scale tests are useful, and have therefore added larger scaling experiments in terms of infoset count and polynomial degree, which show a steep runtime increase and make the practical boundary of the current SOS/SDP implementation much clearer. Due to space limits, we refer to our response to Reviewer GSZ5/the previous scalability comment for the full table and details.

---

> > ### Author Rebuttal · Reviewer_XaMZ · 2026-04-03
> >
> > Thank you for the rebuttal. I prefer to maintain my current score.
> > First, regarding the experimental section (W3), I am still unconvinced of its value and sufficiency in the context of this paper. The authors acknowledge that the experiments merely test global convergence on randomly generated instances. The scale and quantity of these tests are far too limited to demonstrate the practical applicability of the theoretical framework. Even the newly added "absentminded driver" example only relies on only a single core variable p.
> > Second, regarding the theoretical contribution, my primary concern lies in the presentation.  As I originally noted, "the precise mathematical definition of this genericity assumption is somewhat vaguely presented in the main text." Because the authors did not clearly and explicitly define the boundary conditions under which their theory applies within the manuscript, it is difficult for a reader to accurately assess whether the theoretical contribution is as broad and significant as claimed.
> > In conclusion, while I appreciate the clarifications provided in the rebuttal, the inherent limitations of the experimental evaluation and the vagueness of the assumptions in the main text prevent me from raising my score.

---

> > > ### Author Response · Authors · 2026-04-05
> > >
> > > We thank the reviewer for their continued engagement. Below we clarify the two remaining concerns. If the reviewer finds these clarifications helpful, we would be grateful for a reconsideration of the current score.
> > > > Limitation of the experimental evaluation.
> > >
> > > We respectfully emphasize that this is primarily a *theoretical* paper. Accordingly, the experiments are intended to validate the proposed methods on IREFGs, rather than to demonstrate large-scale practical competitiveness. We explicitly acknowledge the current scalability limitations in the experimental section and discuss several promising directions, including DSOS/SDSOS, low-rank SDP methods, and other more tractable solver variants.
> > >
> > > We would also note that imperfect-recall games currently have very few standard computational benchmark families, in part because there have been no available general-purpose exact computational methods. In that sense, defining and solving broader benchmark families driven by canonical game structures is already meaningful in this setting.
> > >
> > > Regarding the simplicity of the new experiment, we can introduce a new family of games based on the absentminded taxi driver game, parametrized by multiple variables. This extends the classic example to games with multiple actions (i.e., multiple road choices for the driver) and multiple infosets. The game has $\ell$ repeated information sets $I_1,\dots,I_\ell$, where infoset $I_j$ is visited $d_j$ times. At each $I_j$, there is one continuation action and $m_j-1$ terminal actions, so the behavioral variable is $x_j=(x_{j,0},x_{j,1},\dots,x_{j,m_j-1})\in\Delta^{m_j-1}$, where $x_{j,0}$ is the continuation probability. If terminal action $a_{j,b}$ is chosen at the $k$-th visit of $I_j$, the payoff is $r_{j,k,b}$, while the always-continue outcome has penalty $r_{\mathrm{term}}=-1$. The induced ex-ante utility is
> > > $$
> > > u(x_1,\dots,x_\ell)=
> > > \sum_{j=1}^{\ell}
> > > \left(\prod_{h<j} x_{h,0}^{d_h}\right)
> > > \left(
> > > \sum_{k=1}^{d_j} x_{j,0}^{k-1}\sum_{b=1}^{m_j-1} r_{j,k,b}x_{j,b}
> > > \right)
> > > +
> > > r_{\mathrm{term}}\prod_{h=1}^{\ell} x_{h,0}^{d_h},
> > > $$
> > > which is a polynomial over $\prod_{j=1}^{\ell}\Delta^{m_j-1}$ of total degree $D=\sum_{j=1}^{\ell} d_j$.
> > >
> > > For a representative case with $\ell=3$, $d=[2,2,2]$, and action-set sizes $m=[2,3,4]$, this yields a degree-$6$ instance with 9 variables. Using the degree-$6$ SOS relaxation, we extracted atomic solutions in $100/100$ random instances, with average runtime $1.56$s. Thus, we are no longer limited to the one-variable absentminded driver case: they cover a broader imperfect-recall class of games with multiple infosets and multi-action sets. We will add this expanded benchmark family in the revision.
> > >
> > > > Vagueness of the genericity assumption.
> > >
> > > We respectfully emphasize that the “measure-zero” notion of genericity is mathematically precise and standard in the literature. However, as the reviewer noted, a measure-zero condition is not easily verifiable, and, as discussed in our response, a canonical practical approach is to show that the exceptional set is contained in a proper algebraic variety.
> > >
> > >
> > > In our setting, after fixing the game structure and admissible utility class, the simplex constraints are fixed and only the utility coefficients $c$ vary. For each active-set pattern $\alpha$ of the nonnegativity constraints, let $F_\alpha(z;c)=0$ be the corresponding polynomial KKT system, where $z$ collects the behavioral variables and multipliers. By standard results in elimination theory [1], there exists a nonzero polynomial $\Delta_\alpha(c)$ such that the exceptional coefficients for this system lie in $\{\Delta_\alpha(c)=0\}$. We therefore call $c$ generic if $\Delta_\alpha(c)\neq 0$ for every $\alpha$. Equivalently, if $\Delta(c):=\prod_\alpha \Delta_\alpha(c)$, then nongeneric instances satisfy $\Delta(c)=0$. Thus the exceptional set is a proper algebraic variety, and hence *by definition* has Lebesgue measure zero.
> > >
> > > Concretely, we describe a simple 2-player example: let
> > > $$u_1(x,y)=\frac{a}{2}x^2+bxy+px,\quad u_2(x,y)=\frac{d}{2}y^2+exy+qy,$$
> > > with coefficient vector $c=(a,b,p,d,e,q)$. For the interior active-set pattern, the KKT system is $$ax+by+p=0,\quad ex+dy+q=0,$$ and the exceptional set is exactly $\Delta_\alpha(c)=ad-be=0$. Thus, under any continuous sampling of the utility coefficients, this exceptional case occurs with probability $0$, and the KKT system has finitely many solutions almost surely.
> > >
> > > We will clearly state the full definition along with examples in the appendix, and write the main text definition more clearly. Operationally, the reviewer seems concerned that the class of games we consider is restrictive in some sense. Again, we clarify that the “measure-zero” statement means that after fixing a game structure (such as in our zero-sum example in the rebuttal), we recover finite convergence *almost surely*.
> > >
> > > *[1] D. Lazard and F. Rouillier. Solving parametric polynomial systems. J. Symbolic Comput., 2007.*

---

### Official Review · Reviewer_uAbR · 2026-03-12

**Soundness:** 3
**Presentation:** 3
**Significance:** 3
**Originality:** 3
**Overall Recommendation:** 5
**Confidence:** 4

**Summary:**

This paper proposes a novel framework for solving Imperfect-Recall Extensive-Form Games (IREFGs) via Sum-of-Squares (SOS) optimization. The paper focuses on non-absentminded IRREFGs (NAM-IREFGs), where players can forget parts of past history. These games are more realistic than perfect-recall EFGs and highly relevant to AI and decision-making domains. The framework computes ex-ante optimal strategies in single-player IREFGs using an SOS hierarchy and extends the approach to multi-player settings to find behavioral Nash equilibria.

* Problem: IREFGs are computationally challenging because imperfect recall prevents standard standard techniques designed for perfect-recall games from directly applying, yet they better capture real-world applications. Existing methods struggle with global optimality, equilibrium computation, and proving non-existence in multi-player cases.

* Solution: The framework reformulates single-player IREFGs as polynomial optimization problems and applies the Moment-SOS hierarchy to obtain approximate optimal strategies. For multi-player IREFGs, it introduces a Select-Verify-Cut (SVC) framework that searches for Nash equilibria or certifies non-existence. This provides the first polynomial-time tractable approach for broad subclasses of IREFGs, overcoming limitations of prior methods.

* Theoretical contribution: The authors extend the approach to multi-player settings via the SVC framework, providing finite convergence for specific subclasses such as NAM-IREFGs. In addition, they define new SOS-Concave and SOS-Monotone game classes that guarantee existence and uniqueness of behavioral Nash equilibria.

* Empirical results: Experiment demonstrate that the proposed approach achieves superior global optimality compared to Projected Gradient Descent (PGD) on single-player IREFGs across 50 random instances.

**Compliance With Llm Reviewing Policy:**

Affirmed.

**Final Justification:**

The rebuttal adequately addressed my primary concerns. I maintain my score and recommend acceptance.

**Key Questions For Authors:**

1. SOS-Concavity / SOS-Monotone Prevalence
> How common are SOS-concave and SOS-monotone games in real-world settings?

2. Baseline Selection
> Why was PGD chosen as the sole empirical baselines, rather than comparing against a broader set of optimization or game-solving methods?

3. Large-Scale AI Relevance
> How do the authors envision this framework contributing to large-scale AI systems in practice, such as those used in AlphaStar- or DeepNash-style setting?

**Limitations:**

Yes.

**Strengths And Weaknesses:**

1. Strength
* Soundness
> * The transformation of IREFGs into polynomial optimization is mathematically rigorous, and the proofs for the proposed theorems are clearly presented.
> * Experiments show that SOS obtain consistent global solutions, providing strong empirical support for the theory.

* Presentation
> * The paper is well-structured, with a logical flow.
> * Equations are clearly presented, and examples such as Figure 1 are helpful for understanding.

* Significance
> * IREFGs represent one of challenges in AI (e.g. large-scale imperfect information games) and decision-making.
> * By identifying tractable subclasses via SOS and providing NE computation/existence, it overcomes limitations of prior methods.

* Originality
> * The idea of extending Momont-SOS to multi-player IREFGs via the SVC framework is original.
> * Defining new SOS-Concave and SOS-Monotone game classes offer a novel perspective.

2. Weakness
* Soundness
> * In the multi-player setting, the number of SVC iteration can grow exponentially with game size, yet a worst case analysis is missing.
> * Additional discussion is needed on how realistic the assumptions (e.g. generic utilities) are for practical games.

* Significance
> * Experiments are conducted only on small-scale instances. Testing on large games (e.g. poker variants) is lacking.
> * Comparisons are limited to PGD.

* Originality
> * While the authors clearly cite the relevant literature, the core idea of using SOS in single-player IREFGs originates from prior work.

---

> ### Author Rebuttal · Authors · 2026-03-31
>
> We are grateful to the reviewer for the support of our paper, and for the detailed and insightful comments. We will respond to each of your comments separately below.
> > W1: Worst-case growth of SVC iterations.
>
> Theorem 5.1(ii) already implies a finite worst-case bound under generic utilities: the number of SVC iterations is at most the number of feasible real KKT candidates, since each failed candidate is permanently cut off without removing any NE. Thus, a worst-case bound is already implicit in our proof, although it is combinatorial/algebraic, and may still grow exponentially with game size. We will clarify this in the revision.
> > W2: Realism of generic utilities.
>
> The definition of genericity in terms of a Lebesgue measure-zero set is standard, see, e.g., [1], and is intended to characterize a property that holds almost everywhere. More concretely, it suffices that the exceptional cases are those for which certain polynomial equalities hold. Such cases form a proper algebraic variety, and every proper algebraic variety has Lebesgue measure zero [2].
>
> In our setting, this guarantees that for a fixed game structure and utility class, the players’ best-response problems have finitely many KKT solutions. For instance, our framework applies to *generic* zero-sum games. Concretely, this means that the first $n-1$ utility functions are chosen outside a measure-zero set while the last is determined by the zero-sum constraint. We will add a discussion on this in subsequent versions.
>
> *[1] J. Nie and K. Ranestad. Algebraic degree of polynomial optimization. SIAM J. Optim., 2009.*
>
> *[2] J. Harris. Algebraic Geometry: A First Course. Springer, 2013.*
> > W3: Experiments on large games.
>
> Our goal is to compute NE globally in a notoriously NP-hard class of games. The main contribution is theoretical, and the experiments are intended mainly to support the claimed global-optimality guarantees. We nevertheless agree that larger-scale tests are useful, and have therefore added larger scaling experiments. Due to space limits, we refer to our response to Reviewer GSZ5/the previous scalability comment for details.
>
> For comments on the choice of PGD as a baseline (W4), see our response to Q2 below.
> > W5: Limited originality in the single-player case.
>
> We agree that prior work covers (i) reducing single-player IREFGs to polynomial optimization and (ii) solving polynomial optimization via Moment-SOS. Our contribution is to explicitly connect these two ingredients into a solver framework for IREFGs, together with convergence and exactness guarantees. To the best of our knowledge, this explicit SOS-based algorithmic treatment of IREFGs has not been developed before.
> > Q1: SOS-Concavity/SOS-Monotone Prevalence.
>
> We do not claim that SOS-concave or SOS-monotone games are universally representative of real-world IREFGs. Rather, they are structured, SDP-checkable subclasses that isolate favorable regimes where the SOS hierarchy becomes much sharper. Their role is analogous to convex/monotone subclasses of continuous games: not universal, but useful for identifying tractable cases with strong guarantees. In particular, concave quadratic utilities are automatically SOS-concave, and a number of structured polynomial interaction models fall into or near these regimes.
> > Q2/W4: Lack of broader baselines.
>
> To our knowledge, there is no other general computational baseline for ex-ante global optima/Nash equilibria in general IREFGs. We therefore used PGD as a ‘local’ baseline since it is known to converge to KKT-type solutions [3] and was used in recent imperfect-recall work focused on local optima [4]. The purpose of the experiments was to contrast our globally certified solutions with a representative method for computing locally optimal equilibria, which is why PGD was chosen. We will clarify this in the revision.
>
> *[3] J. Fearnley et al. The complexity of gradient descent: CLS = PPAD ∩ PLS. J. ACM, 2022.*
>
> *[4] E. Tewolde et al. Decision making under imperfect recall: Algorithms and benchmarks. arXiv, 2026.*
>
> > Q3: Extension to large-scale AI systems.
>
> Regarding large-scale AI systems, we view our framework primarily as a theoretical foundation rather than a directly deployable solver for AlphaStar-/DeepNash-style systems. The main challenge is scalability: both the SDP size and the hierarchy level can grow quickly with problem size. An analogy can be drawn to solving large imperfect-information EFGs with perfect recall, where theoretical guarantees were established long before practical advances such as CFR/MCCFR allowed for meaningful large-scale applications. Beyond improved SDP solvers and tractable SOS variants, a promising direction is sample-based SOS methods that do not require full global relaxations. Recent sampling-based SDP results in control/robotics [5] suggest this may be a viable path.
>
> *[5] A. Groudiev et al. Sampling-based global optimal control and estimation via semidefinite programming. arXiv, 2025.*

---

> > ### Author Rebuttal · Reviewer_uAbR · 2026-04-03
> >
> > Thank you for the comprehensive rebuttal. We have gone through all the responses to W1–W5 and Q1–Q3, and found them well-reasoned and convincing — especially the worst-case bound for SVC iterations, the clarification of generic utilities, the motivation behind the PGD baseline, and the newly added experiments in your response to Reviewer GSZ5. We look forward to seeing the revised version.

---

> > > ### Author Response · Authors · 2026-04-05
> > >
> > > Thank you for the thoughtful follow-up and positive reassessment. We are glad the clarifications and added experiments were helpful, and we will incorporate them carefully in the revision.

---

### Official Review · Reviewer_unMK · 2026-03-13

**Soundness:** 3
**Presentation:** 4
**Significance:** 2
**Originality:** 2
**Overall Recommendation:** 4
**Confidence:** 3

**Summary:**

The paper applies the Moment-SOS hierarchy from polynomial optimization to compute equilibria in imperfect-recall extensive-form games (IREFGs). The authors investigate what guarantees the Moment-SOS hierarchy provides for solving single-player and multi-player IREFGs and identifies structured subclasses where it enables tractable computation. For single-player IREFGs, the authors show that the Moment-SOS hierarchy converges asymptotically to the ex-ante optimal value (and the convergence is finite under a genericity assumption). And if the game is also non-absentminded (NAM), they prove exact convergence at a level of the hierarchy depending on number of infosets. They prove similar results for convergence to Nash Equilibrium in multi-player games. And finally, they define the subclasses of SOS-concave and SOS-monotone IREFGs, where the moment-sos hierarchy converges at the first level for single-player games.

**Compliance With Llm Reviewing Policy:**

Affirmed.

**Final Justification:**

My concerns were addressed by the rebuttal and I have decided to maintain my positive score.

**Key Questions For Authors:**

* How prevalent is the genericity assumption in actual natural classes of games? Eg. how about team games? Do they impose some structure on the payoffs that might fall into the measure-zero exceptions of the genericity? Are there any other natural classes of games that might have this property?

**Limitations:**

yes

**Strengths And Weaknesses:**

# Strengths

* The paper is very cleanly written and the contribution is very naturally motivated.

* The introduction of the vertex-restricted Moment-SOS hierachy to deal with NAM IREFGs is an interesting strong structural simplification. Same for the KKT-based hierarchy.

* The SOS-concave and SOS-monotone subclasses are useful characterizations of games that admit an SDP solution.

* The experimental evaluation is a nice addition.

# Weaknesses

* Overall the paper is a nice contribution to the literature and is a nice merge of games and polynomial optimization for upper bounds, but its theoretical/technical contributions are not very novel beyond that.

* In the experimental results, why do you not compare with MILP-based methods? Also, would it make sense to test low-level SOS relaxations and how they compare to the optimal values of the games?

---

> ### Author Rebuttal · Authors · 2026-03-31
>
> We are grateful to the reviewer for the support of our paper, and for the detailed and insightful comments. We will respond to each of your comments separately below.
> > W1: Limited novelty beyond the connection.
>
> While our framework builds on the SOS optimization literature, our results are not a direct black-box application: we develop several new results that exploit the specific structure of IREFGs and require additional analysis and technical insights.
>
> (a) Classical finite-convergence results for SOS hierarchies assume generic data (both utilities and constraints). In our setting, the feasible region is a product of simplices with highly structured, non-generic constraints. We assume genericity only of the utility and adapt the finite-convergence arguments to this particular game-theoretic geometry.
>
> (b) For single-player non-absentminded IREFGs, we prove convergence at level $\ell+1$, where $\ell$ is the number of infosets. This yields an explicit, game-structural degree bound, going well beyond what generic polynomial optimization theory would provide.
>
> (c) We introduce the subclasses of SOS-concave/SOS-monotone IREFGs, for which the hierarchy converges in one step. These SOS variants are SDP-checkable, so they give a tractable certificate for favorable game structure, whereas verifying concavity or monotonicity directly is generally NP-hard.
>
> > W2: On missing MILP-based comparisons.
>
> We did not include MILP-based baselines because the MILP methods only apply to much narrower IREFG subclasses—specifically, special/non-absentminded two-player zero-sum settings or single-player non-absentminded settings. Our paper instead studies single-player ex-ante global optimization and general multi-player NE computation, including absentminded games. To the best of our knowledge, there is no other general computational baseline for ex-ante global optima or NE in general IREFGs. We therefore chose PGD as a standard local baseline, since it targets stationary/KKT-type solutions and is commonly used in prior imperfect-recall work that focuses on local optima [1]. The purpose of the experiments was to contrast our globally certified solutions with a representative method for computing locally optimal equilibria. We will clarify this point in the revision.
>
> *[1] E. Tewolde et al. Decision making under imperfect recall: Algorithms and benchmarks. arXiv, 2026.*
>
> > W3: Value of low-level SOS relaxations.
>
> This is an interesting question. While we have determined important subclasses where our method provably converges at the *first* level, in general it would be useful to have theoretical upper bounds on the approximation error as a function of the hierarchy level. For Putinar-type hierarchies, recent theory gives worst-case bounds of the form $O(1/r^c)$, where $r$ denotes the relaxation order, and $c>0$ depends on the problem class; see Table 2 in [2] for a concise summary.
>
> *[2] M. Laurent and L. Slot. An overview of convergence rates for sum of squares hierarchies in polynomial optimization. ICIAM, 2023.*
>
> > Q1: Prevalence of the genericity assumption.
>
> The definition of genericity in terms of a Lebesgue measure-zero set is standard, see, e.g., [3], and is intended to characterize a property that holds almost everywhere. A more concrete and practically useful sufficient condition is that the property fails only on a proper algebraic variety, that is, on the solution set of a system of polynomial equations. Indeed, every proper algebraic variety has Lebesgue measure zero [4], and therefore any property that fails only on such a set is generic. For example, the property that a square matrix is invertible is generic, since it fails precisely when its determinant vanishes, and the set of singular matrices is the proper algebraic variety defined by the polynomial equation $\det(A)=0$.
>
> In our setting, the notion of a generic game ensures that the players’ best-response problems have only finitely many KKT solutions, after fixing the game structure and the admissible utility class. For instance, our framework applies to *generic* zero-sum games. Concretely, this means that the first $n-1$ utility functions are chosen outside a measure-zero set (i.e., outside a proper algebraic variety), while the last utility function is then determined by the condition that the utilities sum to zero. Finally, an analogous characterization of genericity holds for team games, since any IREFG can be viewed as a team game where each team is represented by an imperfect-recall player.
>
> *[3] J. Nie and K. Ranestad. Algebraic degree of polynomial optimization. SIAM J. Optim., 2009.*
>
> *[4] J. Harris. Algebraic Geometry: A First Course. Springer, 2013.*

---

> > ### Author Rebuttal · Reviewer_unMK · 2026-04-03
> >
> > Thank you! I maintain my positive evaluation of the paper.

---

> > > ### Author Response · Authors · 2026-04-05
> > >
> > > Thank you for the positive evaluation and for maintaining your support. We appreciate it and will incorporate the clarifications in the revision.

---

### Official Review · Reviewer_GSZ5 · 2026-03-17

**Soundness:** 3
**Presentation:** 3
**Significance:** 3
**Originality:** 3
**Overall Recommendation:** 5
**Confidence:** 3

**Summary:**

The authors investigate the computation of optimal strategies and equilibria in imperfect-recall extensive-form games (IREFGs), a setting where equilibrium computation is known to be provably hard. By exploiting the equivalence between IREFGs and polynomial optimization over products of simplices, the authors apply the Moment-Sum-of-Squares (Moment-SOS) hierarchy to these games. They prove that for single-player IREFGs, the hierarchy converges asymptotically to the ex-ante optimal value, and achieves exact convergence at a finite level for non-absentminded (NAM) games. For multi-player IREFGs, the authors introduce a method to compute behavioral Nash equilibria by adapting the Select-Verify-Cut (SVC) framework. Finally, new subclasses of "SOS-concave" and "SOS-monotone" games where convergence occurs at the very first level of the hierarchy are introduced.

**Compliance With Llm Reviewing Policy:**

Affirmed.

**Final Justification:**

I had only minor comments, and the rebuttal was good. I think this is a good paper, and the "accept" score is fair.

**Key Questions For Authors:**

Can you run on larger domains?

**Limitations:**

Yes

**Strengths And Weaknesses:**

I think this is a solid contribution. The paper seems to be written very carefully; the arguments are clean, and the exposition is mostly clear. I have not even noticed any typos, and I have only very few minor remarks. A substantial portion of the paper is dedicated to definitions and setup. While this is likely unavoidable due to strict conference page limits, providing a bit more intuitive context for the SOS methods themselves would have improved readability. I have not checked the appendices in detail (hence my lower confidence), but the underlying arguments appear sound and logically consistent.

*Main strengths*:
Theoretical Significance in a Hard Domain: The theoretical results seem interesting to me, and represent a rare, positive algorithmic development for a notoriously difficult setting of imperfect-information games. While these results are primarily of theoretical interest and may not immediately give practical algorithms, establishing provable convergence guarantees in a space where even finding a local optimum is NP-hard is, in my opinion, a strong contribution.

Successful Adaptation of SVC: The authors successfully adapt the SVC framework (using previous results in the literature) to the multi-player IREFG setting. This provides a rigorous, concrete method to either find a behavioral Nash equilibrium or formally certify its nonexistence.

*Main weaknesses and limitations*:
Scope of Empirical Evaluation: The experiments are conducted on very small games. Given that the reported running times for these instances range only from 0.01 seconds to about 9 seconds, the authors could have easily run experiments on larger instances to better demonstrate the empirical limits and practical relevance of their approach.
Scalability and Computational Tax: The size of the Semidefinite Programs (SDPs) required for the SOS hierarchies scales poorly, which limits the practical applicability of the method.  The computational tax for achieving SOS-certified global optimality is thus quite high, especially when compared to the probabilistic baseline methods.

*Specific (minor) comments*:
- Sometimes the definitions are emphasized and sometimes not. This should be made consistent.
page 3, column 2, lines 147-148: The definition of ex-ante optimal behavioral strategy is not very clear, as I believe "u" was not properly introduced.
- Some accents in the references are broken or missing, for example, in the names Lisý or Bošanský.

---

> ### Author Rebuttal · Authors · 2026-03-31
>
> We are grateful to the reviewer for the support of our paper, and for the detailed and insightful comments. We will respond to each of your comments separately below.
> > W1/Q1: Empirical scale and larger domains.
>
> We wish to clarify that our goal is to obtain global Nash equilibria in a notoriously NP-hard class of games. Our primary contribution is theoretical, and the empirical results we present are intended to corroborate the claimed global optimality of our method. However, we agree with the reviewer that larger-scale experiments would be useful, particularly to provide adequate information for practitioners who might wish to implement our techniques.
>
> To address this, we added larger single-player IREFG instances with 3 actions per information set, varying both the number of information sets $\ell$ and the polynomial utility degree $d_u$. We also improved the implementation by using the simplex equality constraints to eliminate redundant variables before constructing the SOS relaxation, which reduces the problem size. The table below reports average runtime (seconds); in all reported cases, SOS attained the optimum at the relaxation degree matching $d_u$. We imposed a 20-minute maximum time budget per experiment. Entries marked “--” denote settings we did not pursue further once the last reported instance in that row or column had already approached this limit and made the scaling boundary clear.
>
> The new results show that larger domains are indeed solvable, but with rapidly increasing cost. At fixed $d_u=6$, runtime grows from $0.086$s at $\ell=2$ to $1694.92$s at $\ell=6$. At fixed $\ell=2$, increasing $d_u$ from $4$ to $16$ raises runtime from $0.018$s to $1674.72$s. We will include these expanded results in the revision to better document the empirical limit and practical relevance of the proposed SOS/SDP method.
>
> |$\ell \backslash d_u$|4|6|8|10|12|14|16|
> |---|---|---|---|---|---|---|---|
> |2|0.018|0.086|0.73|6.64|49.74|321.64|1674.72|
> |3|0.049|1.62|54.27|1386.38|--|--|--|
> |4|0.22|25.05|2084.43|--|--|--|--|
> |5|0.88|231.80|--|--|--|--|--|
> |6|3.80|1694.92|--|--|--|--|--|
> |7|20.90|--|--|--|--|--|--|
> |8|103.92|--|--|--|--|--|--|
> |9|2668.71|--|--|--|--|--|--|
>
> > W2: Scalability and computational tax.
>
> We agree that SOS hierarchies can incur a substantial computational cost, and we do not claim superior scalability over local baselines. Our goal is instead to provide a globally certified framework for IREFGs; intuitively, global certification is more demanding than computing only a local or stationary solution. At the same time, to alleviate some concerns on scalability, we identify important regimes where this cost drops sharply. For single-player non-absentminded IREFGs, exact convergence is guaranteed with one SDP at an explicit level $\ell+1$, where $\ell$ is the number of infosets. For single-player SOS-concave/SOS-monotone IREFGs, exactness already holds with one SDP at the first admissible relaxation level, determined by the game degree. We also explicitly acknowledge the broader scalability challenge in the experimental section and discuss several promising directions, including DSOS/SDSOS, low-rank SDP methods, and other more tractable solver variants. Exploring such implementations to improve scalability is an important direction for future work.
>
> > W3: Consistency of presentation and clarity of definition.
>
> We will standardize the formatting of definitions throughout the paper to improve consistency. Regarding the notation $u$, we note that $u_i(\mu)$ is defined on page 3, column 2, line 141 as the expected utility of player $i$ under behavioral strategy profile $\mu$. We also state on page 3, column 2, line 138 that, in the single-player case, we set $n=1$ and drop the player index $i$ for clarity. Thus, $u(\mu)$ is introduced as the single-player shorthand for $u_i(\mu)$.
>
> > W4: Reference formatting.
>
> We will proofread the bibliography and fix broken or missing diacritics in the references.

---

> > ### Author Rebuttal · Reviewer_GSZ5 · 2026-04-03
> >
> > Thank you for your answers.
> > I think this is a good paper, and the "accept" score is fair.

---

> > > ### Author Response · Authors · 2026-04-05
> > >
> > > Thank you for the positive reassessment. We are glad the rebuttal helped clarify the paper, and we will incorporate these clarifications in the revision.

---

### Decision · Program_Chairs · 2026-04-30

**Decision:**

Accept (regular)

**Comment:**

The paper addresses equilibrium computation in extensive-form games with imperfect recall, a setting that is notoriously hard and rarely tackled in the literature on algorithmic game theory. The paper proposes sum-of-squares (SOS) hierarchies for computing Nash equilibria, working over behavioral strategies. It shows that (i) these hierarchies converge asymptotically, (ii) under genericity assumptions, the convergence is finite, and (iii) in single-player non-absentminded games, convergence occurs at a finite level. Finally, the paper also introduces the new classes of (SOS)-concave and (SOS)-monotone games, and show that in the single-player setting the SOS hierarchy converges at the first level, enabling equilibrium computation with a single semidefinite program.

After the Authors' rebuttals, all the Reviewers agree that the paper should be accepted at the conference, and, thus, I will suggest acceptance of the paper. The only (minor in my opinion) concerns that remain are on the presentation quality and the limited experimental evaluation. I strongly encourage the Authors to address all the Reviewers' comments in the final version of the paper, and possibly expand the experimental evaluation, maybe by enlarging the testbed of games.